

**Organic peroxy radical chemistry in oxidation flow reactors and environmental chambers**
**and their atmospheric relevance**
Zhe Peng[1], Julia Lee-Taylor[1,2], John J. Orlando[2], Geoffrey S. Tyndall[2] and Jose L. Jimenez[1]
[1] Cooperative Institute for Research in Environmental Sciences and Department of Chemistry,
University of Colorado, Boulder, Colorado 80309, USA
[2] Atmospheric Chemistry Observation and Modeling Laboratory, National Center for Atmospheric
Research, Boulder, Colorado 80307, USA
Correspondence: Zhe Peng (zhe.peng@colorado.edu) and Jose L. Jimenez
(jose.jimenez@colorado.edu)
**Abstract.** Oxidation flow reactors (OFR) are a promising complement to environmental chambers for
investigating atmospheric oxidation processes and secondary aerosol formation. However, questions
have been raised about how representative the chemistry within OFRs is of that in the troposphere. We
investigate the fates of organic peroxy radicals ($RO_2$), which play a central role in atmospheric organic
chemistry, in OFRs and environmental chambers by chemical kinetic modeling, and compare to a variety
of ambient conditions to help define a range of atmospherically relevant OFR operating conditions. For
most types of $RO_2$, their bimolecular fates in OFRs are mainly $RO_2+HO_2$ and $RO_2+NO$, similar to chambers
and atmospheric studies. For substituted primary $RO_2$ and acyl $RO_2$, $RO_2+RO_2$ can make a significant
contribution to the fate of $RO_2$ in OFRs, chambers and the atmosphere, but $RO_2+RO_2$ in OFRs is in general
somewhat less important than in the atmosphere. At high NO, $RO_2+NO$ dominates $RO_2$ fate in OFRs, as
in the atmosphere. At high UV lamp setting in OFRs, $RO_2+OH$ can be a major $RO_2$ fate and $RO_2$
isomerization can be negligible for common multifunctional $RO_2$, both of which deviate from common
atmospheric conditions. In the OFR254 operation mode (where OH is generated only from photolysis
of added $O_3$), we cannot identify any conditions that can simultaneously avoid significant organic
photolysis at 254 nm and lead to $RO_2$ lifetimes long enough (~10 s) to allow atmospherically relevant
$RO_2$ isomerization. In the OFR185 mode (where OH is generated from reactions initiated by 185 nm
photons), high relative humidity, low UV intensity and low precursor concentrations are recommended
for atmospherically relevant gas-phase chemistry of both stable species and $RO_2$. These conditions
ensure minor or negligible $RO_2+OH$ and a relative importance of $RO_2$ isomerization in $RO_2$ fate in OFRs
within ~x2 of that in the atmosphere. Under these conditions, the photochemical age within OFR185
systems can reach a few equivalent days at most, encompassing the typical ages for maximum
secondary organic aerosol (SOA) production. A small increase in OFR temperature may allow the relative
importance of $RO_2$ isomerization to approach the ambient values. To study heterogeneous oxidation of
SOA formed under atmospherically-relevant OFR conditions, a different UV source with higher intensity
is needed after the SOA formation stage, which can be done with another reactor in series. Finally, we
recommend evaluating the atmospheric relevance of $RO_2$ chemistry by always reporting measured
and/or estimated OH, $HO_2$, NO, $NO_2$ and OH reactivity (or at least precursor composition and
concentration) in all chamber and flow reactor experiments. An easy-to-use $RO_2$ fate estimator program
is included with this paper to facilitate investigation of this topic in future studies.



## 1   Introduction

Laboratory reactors are needed to isolate and study atmospheric chemical systems. Environmental
chambers have been a major atmospheric chemistry research tool for decades (Cocker et al., 2001;
Carter et al., 2005; Presto et al., 2005; Wang et al., 2011; Platt et al., 2013). Over the last few years,
oxidation flow reactors (OFRs) (Kang et al., 2007) have emerged as a promising complement to
chambers, and are being used to investigate atmospheric oxidation processes, particularly volatile
organic compound (VOC) oxidation and secondary organic aerosol (SOA) formation and aging (Kang et
al., 2011; Lambe et al., 2015; Hu et al., 2016; Palm et al., 2016). These processes have air quality (Levy
II, 1971), human health (Nel, 2005) and climate impacts (Stocker et al., 2014).
The most important advantage of OFRs is their ability to achieve relatively high photochemical
ages (on the order of equivalent hours or days (assuming an average ambient OH concentration of
$1.5 \times 10^6$ molecules cm$^{-3}$; Mao et al., 2009) in minutes instead of hours in chambers (Lambe et al., 2011).
Rapid aging is usually achieved by highly active HO$_x$ radical chemistry initiated by low-pressure Hg lamp
emissions (185 and 254 nm) (Li et al., 2015; Peng et al., 2015). This allows shorter residence times in
OFRs thus reducing gas and particle losses to walls, which can be very important in Teflon chambers
(Cocker et al., 2001; Matsunaga and Ziemann, 2010; Zhang et al., 2014; Krechmer et al., 2016). In
addition, lower costs and small size (volumes of the order of 10 L) of OFRs allow better portability. These,
together with the ability to rapidly achieve high photochemical ages, are advantageous for field
applications.
These advantages of OFRs have led a number of atmospheric chemistry research groups (Lambe
and Jimenez, 2018) to deploy them in field (Hu et al., 2016; Ortega et al., 2016; Palm et al., 2016, 2017),
source (Ortega et al., 2013; Tkacik et al., 2014; Karjalainen et al., 2016; Link et al., 2016) and laboratory
studies (Kang et al., 2011; Lambe et al., 2013; Richards-Henderson et al., 2016; Lim et al., 2017).
However, the atmospheric relevance of VOC oxidation and SOA formation simulated in OFRs has
repeatedly been called into question, because the UV wavelengths most commonly used to initiate OFR
chemistry do not exist in the troposphere, and because OH levels in OFRs ($10^8$–$10^{10}$ molecules cm$^{-3}$) can
be much higher than tropospheric levels ($10^6$–$10^7$ molecules cm$^{-3}$; Mao et al., 2009; Stone et al., 2012)
To clarify this issue, a series of chemical kinetic modeling studies have been performed: Li et al. (2015)
and Peng et al. (2015) established a radical chemistry and oxidation model whose predictions compare
well against laboratory experiments and found that OH can be substantially suppressed by external OH
reactants (e.g. SO$_2$, NO$_x$ and VOCs externally introduced into the reactor); Peng et al. (2016) identified
low water mixing ratio (H$_2$O) and/or high external OH reactivity (OHR$_{ext}$, i.e. first-order OH loss rate
constant contributed by external OH reactants) as conditions that can cause significant non-
tropospheric VOC reactions (e.g. through photolysis at 185 and/or 254 nm); Peng and Jimenez (2017)
studied NO$_y$ chemistry in OFRs and showed that high-NO conditions, where organic peroxy radicals react
more rapidly with NO than with HO$_2$, can only be realized by simple NO injection in a very narrow range
of physical conditions, whose application to investigating intermediate- and high-NO environments (e.g.
urban area) is limited; Peng et al. (2018) thus evaluated a few new techniques to maintain high-NO





conditions in OFRs and found injection of percent-level $N_2O$ effective to achieve this goal.
While $HO_x$ and $NO_y$ chemistries have been extensively characterized in OFRs so far, organic peroxy
radical ($RO_2$) chemistry has yet to be considered in detail, as previous studies have only considered the
balance between $RO_2$+NO vs $RO_2$+$HO_2$. There has been some speculation that due to high OH
concentrations in OFRs, $RO_2$ concentration and lifetime might be significantly different from ambient
values, leading to dominance of $RO_2$ self/cross reactions and elimination of $RO_2$ isomerization pathways
(Crounse et al., 2013; Praske et al., 2018). Given the central role $RO_2$ plays in atmospheric chemistry
(Orlando and Tyndall, 2012; Ziemann and Atkinson, 2012) and the rapidly increasing use of OFRs, $RO_2$
chemistry in OFRs needs to be studied in detail to characterize the similarities and differences between
their reactions conditions and those in the ambient atmosphere and traditional atmospheric reaction
chambers.
In this paper, we address this need via modeling. All major known fates of $RO_2$ in OFRs will be
investigated and compared with those in typical chamber cases and in the atmosphere. This comparison
will provide insights into the atmospheric relevance of $RO_2$ chemistry in atmospheric simulation reactors
and allow the selection of experimental conditions with atmospherically relevant $RO_2$ chemistry in
experimental planning.
**2    Methods**
Due to a variety of loss pathways of $RO_2$ and a myriad of $RO_2$ types, $RO_2$ chemistry is of enormous
complexity. We detail the $RO_2$ production and loss pathways of interest in this study, the approximations
used to simplify this complex problem, and steps to investigate it methodically. We briefly introduce the
base OFR design and the model, which are described in detail elsewhere (Kang et al., 2007; Peng et al.,

2015, 2018).

**2.1    Potential Aerosol Mass oxidation flow reactor (PAM OFR)**
The concept of the base OFR design simulated in this study, the Potential Aerosol Mass (PAM)
reactor, was first introduced by Kang et al. (2007) The geometry of the most popular PAM OFR is a
cylinder of ~13 L volume. The PAM reactor we simulate is equipped with low-pressure Hg lamps (model
no. 82-9304-03, BHK Inc.) emitting UV light at 185 and 254 nm. When both 185 and 254 nm photons
are used to generate OH (termed "OFR185"), water vapor photolysis at 185 nm produces OH and $HO_2$.
Recombination of $O_2$ and $O(^3P)$, formed by $O_2$ photolysis at 185 nm, generates $O_3$. $O(^1D)$, formed through
$O_3$ photolysis at 254 nm, reacts with water vapor and produces additional OH. 185 nm photons can be
filtered by installing quartz sleeves around the lamps. This converts the reactor into "OFR254" mode,
where photolysis of $O_3$, which must be initially injected, is the only OH production route. The notation
"OFR254-X" is used to specify the initial amount of injected $O_3$ (X ppm) in OFR254. Lambe et al. (2017)
and Peng et al. (2018) have shown that initial injection of $N_2O$ is able to maintain up to tens of ppb NO
in both OFR185 and OFR254. These modes are denoted "OFR185-i$N_2O$" and "OFR254-X-i$N_2O$", or more
generally "OFR-i$N_2O$". In OFR254-i$N_2O$, $O(^1D)$ generated from $O_3$ photolysis reacts with $N_2O$ to generate
NO, while in OFR185-i$N_2O$, $O(^1D)$ is mainly supplied by $N_2O$ photolysis at 185 nm (Peng et al., 2018).
**2.2    $RO_2$ production and loss pathways**



A single generic $RO_2$ is adopted for modeling purposes, to avoid the huge number of $RO_2$ types
that would complicate effective modeling and analysis. In OH-initiated VOC oxidation, $RO_2$ is primarily
produced via VOC+OH → R (+$H_2O$) followed by R+$O_2$ → $RO_2$, where R is hydrocarbyl or oxygenated
hydrocarbyl radical. Since the second step is extremely fast in air (Atkinson and Arey, 2003), the first
step controls the $RO_2$ production rate, which depends on OH concentration and $OHR_{ext}$ due to VOCs
($OHR_{VOC}$, see Appendix A for details). $OHR_{VOC}$ also includes the contribution from oxidation
intermediates of primary VOCs (e.g. methyl vinyl ketone and pinonic acid). When the information about
oxidation intermediates is insufficient to calculate $OHR_{VOC}$, OHR due to primary VOCs is used instead as
an approximant.
Table 1 lists all known $RO_2$ loss pathways. Among those, $RO_2$ photolysis, $RO_2$+$NO_3$ and $RO_2$+$O_3$
are not included in this study, since they are minor or negligible in OH-dominated atmospheres,
chambers and OFRs for the following reasons.
-    The first-order $RO_2$ photolysis rate constant is of the order of $10^{-2}$ $s^{-1}$ at the highest lamp setting in

OFRs (Kalafut-Pettibone et al., 2013) and of the order of $10^{-5}$ $s^{-1}$ in the troposphere under the

assumption of unity quantum yield (Klems et al., 2015), while $RO_2$ reacts with $HO_2$ at >1 $s^{-1}$ at the

highest lamp setting in OFRs and at ~$2 \times 10^{-3}$ $s^{-1}$ in the troposphere. Note that in this study we assume

an average ambient $HO_2$ concentration of $1.5 \times 10^{8}$ molecules $cm^{-3}$ (Mao et al., 2009; Stone et al.,

2012) and $RO_2$+$HO_2$ rate constant of $1.5 \times 10^{-11}$ $cm^3$ $molecule^{-1}$ $s^{-1}$ (Orlando and Tyndall, 2012).

-    When daytime photochemistry is active, $NO_3$ is negligible in the atmosphere. In OFR-i$N_2O$ modes,

$RO_2$+$NO_3$ is negligible unless at very low $H_2O$ and high UV intensity (abbr. UV hereafter), which

result in high $O_3$ to oxidize $NO_2$ to $NO_3$ and keep $HO_2$ minimized. However, very low $H_2O$ causes

serious non-tropospheric organic photolysis (Peng et al., 2016) and thus these conditions are of no

experimental interest.

-    In the atmosphere $RO_2$+$O_3$ is thought to play some role only at night (Orlando and Tyndall, 2012).

Similar conditions may exist in some OFR254 cases, if a very large amount of $O_3$ is injected and $H_2O$

and UV are kept very low to limit $HO_x$ production. These conditions are obviously not OH-

dominated and not further investigated in this study.

Of the $RO_2$ fates considered in this study, $RO_2$+$HO_2$ and $RO_2$+NO and $RO_2$+$RO_2$ have long been
known to play a role in the atmosphere (Orlando and Tyndall, 2012). Despite some small dependencies
on the type of $RO_2$, recommended general rate constants are available for $RO_2$+$HO_2$ and $RO_2$+NO
(Ziemann and Atkinson, 2012; Table 1). We use these recommended values for generic $RO_2$ in this study.
However, $RO_2$ self-/cross-reaction rate constants are highly dependent on the specific $RO_2$ types and can
vary over a very large range ($10^{-17}$–$10^{-10}$ $cm^3$ $molecule^{-1}$ $s^{-1}$). Unsubstituted primary, secondary and
tertiary $RO_2$ self-react at ~$10^{-13}$, ~$10^{-15}$ and ~$10^{-17}$ $cm^3$ $molecule^{-1}$ $s^{-1}$, respectively (Ziemann and Atkinson,
2012). Rate constants of cross-reactions between these $RO_2$ types also span this range (Orlando and
Tyndall, 2012). Substituted $RO_2$s have higher self-/cross-reaction rate constants (Orlando and Tyndall,
2012). $RO_2$+$RO_2$ of highly substituted primary $RO_2$ can be as high as ~$10^{-11}$ $cm^3$ $molecule^{-1}$ $s^{-1}$ (Orlando
and Tyndall, 2012). Very recently, a few highly oxidized 1,3,5-trimethylbenzene-derived $RO_2$s were





reported to self-/cross-react at ~$10^{-10}$ cm$^3$ molecule$^{-1}$ s$^{-1}$ (Berndt et al., 2018). In the present work, we
make a simplification to adapt to the generic RO$_2$ treatment by assuming a single self-/cross-reaction
rate constant for generic RO$_2$ in each case. Three levels of RO$_2$+RO$_2$ rate constants, i.e. $1\times10^{-13}$, $1\times10^{-11}$,
and $1\times10^{-10}$ cm$^3$ molecule$^{-1}$ s$^{-1}$, are studied in this paper. The first level is referred to as "medium RO$_2$+RO$_2$"
as many other RO$_2$ can have self-/cross-reaction rate constants as low as $10^{-17}$ cm$^3$ molecule$^{-1}$ s$^{-1}$; the
second level is defined as "fast RO$_2$+RO$_2$"; the last level is called "very fast RO$_2$+RO$_2$." No RO$_2$+RO$_2$ rate
constant lower than the medium level is investigated in the current work, although there are still a large
variety of RO$_2$ whose self-/cross reactions are at lower rate constants, since at the medium level,
RO$_2$+RO$_2$ is already negligible in all the environments studied in this work, i.e. OFRs, chambers and the
atmosphere (see Section 3.1.1). Since there are only a few very specific examples for very fast RO$_2$+RO$_2$
reported to date, we will not systematically explore this category but compare very fast RO$_2$+RO$_2$ as a
sensitivity case with the other two types of RO$_2$+RO$_2$ reactions.
Acyl RO$_2$ is considered as a separate RO$_2$ type (neither medium nor fast RO$_2$+RO$_2$) in this study
since its reaction with NO$_2$ can be a major sink of RO$_2$ in OFR (Peng and Jimenez, 2017). Thermal
decomposition lifetimes of the product of RO$_2$+NO$_2$, i.e. acylperoxy nitrates, can be hours at laboratory
temperatures (Orlando and Tyndall, 2012; also taken into account in the current work, see Table 1),
while OFR residence times are typically minutes. Besides, acyl RO$_2$ react with many RO$_2$ at ~$10^{-11}$ cm$^3$
molecule$^{-1}$ s$^{-1}$ (Orlando and Tyndall, 2012), similar to that of fast RO$_2$+RO$_2$. We thus assume acyl RO$_2$
self-/cross-reaction rate constant to be also $1\times10^{-11}$ cm$^3$ molecule$^{-1}$ s$^{-1}$ to facilitate the comparison with
fast RO$_2$+RO$_2$ results.
In OFRs operated at room temperature, acylperoxy nitrates barely decompose, while peroxy
nitrates of non-acyl RO$_2$ do decompose on a timescale of 0.1 s (Table 1). As a consequence, the
production and decomposition of peroxy nitrates of non-acyl RO$_2$ reach a steady state in OFRs, which
can be greatly shifted toward the peroxy nitrate side in cases with very high NO$_2$ (Peng and Jimenez,
2017; Peng et al., 2018).
RO$_2$+OH (Fittschen et al., 2014) and RO$_2$ isomerization (Crounse et al., 2013) have recently been
identified as possible significant RO$_2$ fates in the atmosphere. Reactions of the former type, according
to several recent experimental and theoretical studies (Bossolasco et al., 2014; Assaf et al., 2016, 2017b,
2017a; Müller et al., 2016; Yan et al., 2016), have similar rate constants (~$1\times10^{-10}$ cm$^3$ molecule$^{-1}$ s$^{-1}$)
regardless of RO$_2$ type. Therefore, the reaction rate constant of generic RO$_2$ with OH is assigned as $1\times10^{-10}$
$^{-10}$ cm$^3$ molecule$^{-1}$ s$^{-1}$. RO$_2$ isomerization reactivity is highly structure-dependent (Crounse et al., 2013;
Praske et al., 2018) and rate constant measurements are still scarce, preventing us from assigning a
generic RO$_2$ isomerization rate constant. However, for *generic* RO$_2$, isomerization is generally *not* a sink
but a conversion between two RO$_2$ (both encompassed by the generic one in this study), as RO$_2$
isomerization usually generates an oxygenated hydrocarbyl radical, which rapidly recombines with O$_2$
and forms another RO$_2$. Therefore, RO$_2$ isomerization is not explicitly taken into account in the modeling,
but is considered in the RO$_2$ fate analysis.
In summary, 6 pathways are included in the RO$_2$ fate analysis of this study. The need to explore





these 6 pathways for a high number of OFR, chamber, and atmospheric conditions makes presentation
of results challenging. For clarity, we present the results in two steps. In the first step, only well-known
RO$_2$ fates (reaction with NO$_2$, HO$_2$, NO and RO$_2$) will be included in the model. In the second step, the
results of the first step will be used to guide the modeling and analysis of a more comprehensive set of
significant RO$_2$ fates.
**2.3     Model description**

The model used in the present work is a standard chemical kinetic box model, implemented in the

KinSim 3.4 solver in Igor Pro 7 (WaveMetrics, Lake Oswego, Oregon, USA), and has been described in
detail elsewhere (Peng et al., 2015, 2018). Plug flow in the reactor with a residence time of 180 s is
assumed, since the effects of non-plug flow are major only in a narrow range of conditions of little
experimental interest and the implementation of laminar flow or measured residence time distribution
substantially increases computational cost (Peng et al., 2015; Peng and Jimenez, 2017). The reactions
of RO$_2$ discussed in Section 2.2 are added to the chemical mechanism. A generic slow-reacting VOC
(with the same OH rate constant as SO$_2$) is used as the external OH reactant. This slow rate also
represents the generation and consumption of latter-generation products that continue to react with
OH. The reason for this approximation has been discussed in detail in previous OFR modeling papers
(Peng and Jimenez, 2017; Peng et al., 2018). We exclude NO$_y$ species, which are explicitly modeled, from
the calculation of OHR$_{ext}$; thus OHR$_{ext}$ only includes non-NO$_y$ OHR$_{ext}$ hereafter. As OHR$_{ext}$ is dominated
by OHR$_{VOC}$ in most OFR experiments, we use OHR$_{ext}$ to denote OHR$_{VOC}$ in OFRs (while for ambient and
chamber cases OHR$_{VOC}$ is still used to exclude the contribution of CO etc.). The model was estimated to
achieve an accuracy of a factor of 2–3 when compared to field OFR experiments; better agreement can
generally be obtained for laboratory OFR experiments (Li et al., 2015; Peng et al., 2015).

Another key parameter in the model is the HO$_x$ recycling ratio (β), defined in this study as the

number of HO$_2$ molecule(s) produced per OH molecule destroyed by external OH reactants (Peng et al.,
2015). This ratio depends on the products of RO$_2$ loss pathways. The main product of RO$_2$+HO$_2$ is usually
ROOH (Table 1), yielding no recycled HO$_2$, while the main products of RO$_2$+NO are RO and NO$_2$, the
former of which can often undergo extremely fast H-abstraction by O$_2$ to form a carbonyl and HO$_2$. From
VOC oxidation simulations by the fully explicit model GECKO-A (Aumont et al., 2005), we estimate β~0.3
in zero-NO OFRs. At the other extreme, where RO$_2$ is solely consumed by RO$_2$+NO, the product RO yields
HO$_2$ at a branching ratio close to 1, β~1. For intermediate cases, we assume that β may be interpolated
as a linear function of r(RO$_2$+NO)/[r(RO$_2$+NO)+r(RO$_2$+HO$_2$)], where r(RO$_2$+NO) and r(RO$_2$+HO$_2$) are the
local reactive fluxes of RO$_2$+NO and RO$_2$+HO$_2$.

In the present work, we model OFR185, OFR254-70, and OFR254-7 (including their -iN$_2$O variants).

We specify the same temperature and atmospheric pressure (295 K and 835 mbar, typical values in
Boulder, Colorado, USA) as our previous OFR modeling studies (Li et al., 2015; Peng et al., 2015, 2016,
2018; Peng and Jimenez, 2017). The explored physical condition space follows that of our previous OFR-
iN$_2$O modeling work (Peng et al., 2018). The only differences are that in this study we also include cases
without any N$_2$O injected (OFR185 and OFR254 only) and exclude OHR$_{ext}$=0 conditions, which produce





no $RO_2$. In detail, the explored physical condition space covers: $H_2O$ of 0.07–2.3% (relative humidity of
2–71% at 295 K); UV photon flux at 185 nm (abbr. F185) of $1.0 \times 10^{11}$–$1.0 \times 10^{14}$ photons $cm^{-2}$ $s^{-1}$
[corresponding photon flux at 254 nm (F254) of $4.2 \times 10^{13}$–$8.5 \times 10^{15}$ photons $cm^{-2}$ $s^{-1}$]; $OHR_{ext}$ of 1–1000
$s^{-1}$; $N_2O$ mixing ratio (abbr. $N_2O$ hereafter) of 0 and 0.02–20%. All model cases are logarithmically evenly
distributed except for $N_2O$=0 and F254. The latter is calculated based on the F185–F254 relationship for
the lamps simulated here (Li et al., 2015).

For the classification of conditions, the same criteria as in the OFR-i$N_2O$ modeling study (Peng et

al., 2018) are adopted. In detail, high- and low-NO conditions are classified by $r(RO_2+NO)/r(RO_2+HO_2)$.
In the current work, these reactive fluxes are explicitly tracked in the modeling instead of approximated
as in previous studies (Peng and Jimenez, 2017; Peng et al., 2018). The terms "good," "risky" and "bad"
are used to describe OFR operating conditions in terms of non-tropospheric organic photolysis, and are
defined based on the ratios of F185 and F254 exposure ($F185_{exp}$ and $F254_{exp}$, i.e. integrated photon
fluxes over residence time) to OH exposure ($OH_{exp}$), as presented previously (Peng and Jimenez, 2017;
Peng et al., 2018). Briefly, under a given condition non-tropospheric photolysis is of different relative
importance in the fate of each specific organic species: under good conditions, photolysis at 185 and/or
254 nm is unimportant for almost all VOCs; under bad conditions, non-tropospheric photolysis is
problematic for most VOC precursors, since significant photolysis of their oxidation intermediates at
185 and/or 254 nm is almost inevitable; and risky conditions can be problematic for some but not all
VOCs. Note that good/risky/bad conditions refer only to non-tropospheric organic photolysis and *not* to
whether $RO_2$ chemistry is atmospherically relevant. Table S1 summarizes our condition classification
criteria.
**3    Results and discussion**

In this section, the results are presented in two parts, i.e. first for the simulations with well-known

pathways only, and secondly with all significant pathways, as proposed in Section 2.2. Then based on
the results and their comparison with the atmosphere and chamber experiments, we propose
guidelines for OFR operation to ensure atmospherically relevant $RO_2$ chemistry, as well as other
chemistries already discussed in the previous studies (Peng et al., 2016, 2018), in OFRs.
**3.1    Simulations with well-known pathways ($RO_2$+$HO_2$, $RO_2$+$RO_2$, $RO_2$+NO and $RO_2$+$NO_2$)**

Due to significantly different reactivities of non-acyl and acyl $RO_2$, the results of these two types

of $RO_2$ are shown separately.
**3.1.1    Non-acyl $RO_2$**

In this case non-acyl $RO_2$ have only three fates, i.e. $RO_2$+$HO_2$, $RO_2$+NO and $RO_2$+$RO_2$. The relative

importance of these three fates can be shown in a triangle plot (Figure 1). The figure includes data
points of OFR185 (including OFR185-i$N_2O$) and OFR254-70 (including OFR254-70-i$N_2O$), as well as
several typical ambient and chamber studies (Ryerson et al., 2013; Nguyen et al., 2014; Ortega et al.,
2014; Martin et al., 2016, 2017; Carlton et al., 2018; Wofsy et al., 2018). Conditions from the FIXCIT
campaign (Nguyen et al., 2014) are used to represent chamber studies as they were designed for specific
$RO_2$ fates within the limitations of current high-quality laboratory chambers (Table 2). Bad conditions



(in terms of non-tropospheric photolysis) are not shown on these plots because of the lack of
experimental interest. The triangle plots for OFR254-7 (including OFR254-7-iN$_2$O) in the same form
(Figure S1a,b) show no qualitative differences from the results of OFR254-70, implying that initial O$_3$ in
OFR254 modes has only minor impacts on RO$_2$ fate. We see this result not only for well-known non-acyl
RO$_2$ fate, but also for the aspects discussed in the following sections. The similarity between OFR254
modes can be explained by the minor effects of a lower O$_3$ on HO$_x$ at relatively low OHR$_{ext}$ (Peng et al.,
2015). Cases at higher OHR$_{ext}$ often have stronger non-tropospheric photolysis (Peng et al., 2016) and
hence are more likely to be under bad conditions and are not shown in Figs. 1 and S1a,b. For simplicity,
this similarity is not discussed further.

An important feature confirmed in Fig. 1 is that OFR-iN$_2$O modes effectively realize conditions of

experimental interest with variable relative importance of RO$_2$+NO in RO$_2$ fate (Lambe et al., 2017; Peng
et al., 2018). Tuning initially injected N$_2$O can achieve this goal (Fig. 2). While it is possible to reduce
RO$_2$+HO$_2$ in OFR185-iN$_2$O to negligible compared to RO$_2$+NO by increasing N$_2$O, this is not possible in
OFR254-70-iN$_2$O due to fast NO oxidation by the large amounts of O$_3$ added in the reactor. Nevertheless,
OFR254-70-iN$_2$O can still make RO$_2$+NO dominate over RO$_2$+HO$_2$ in RO$_2$ fate. OFR and chamber cases
span a range of ~0–~100% in relative importance of RO$_2$+NO in RO$_2$ fate (Fig. 2), suggesting that both
chambers and OFRs are able to ensure the atmospheric relevance of RO$_2$+NO in RO$_2$ fate.

Another important feature that can be easily seen in Fig. 1 is that medium rate RO$_2$+RO$_2$ (and

hence also RO$_2$+RO$_2$ slower than $10^{-13}$ cm$^3$ molecule$^{-1}$ s$^{-1}$) are of negligible importance in the fate of RO$_2$
(Fig. 1a,c) in OFR185 (including OFR185-iN$_2$O), OFR254-70 (under most conditions, including OFR254-
70-iN$_2$O), chambers and the atmosphere. Thus, a very large subset of RO$_2$ have only minor or negligible
contribution from RO$_2$+RO$_2$ to their fate. This is already known for ambient RO$_2$ fate (Ziemann and
Atkinson, 2012). The reason why this is also true in OFRs is that while OH is much higher than ambient
levels, HO$_2$ and NO (high-NO conditions only) are also higher. One can easily verify that steady-state RO$_2$
concentrations (see Appendix A for details) would not deviate from ambient levels by orders of
magnitude. The reactive fluxes of RO$_2$+RO$_2$ in OFRs are thus not substantially different than in the
atmosphere, while RO$_2$+HO$_2$ and RO$_2$+NO (high-NO conditions only) are both faster in OFRs because of
higher HO$_2$ and NO. The combined effect is a *reduced* relative importance of RO$_2$+RO$_2$ in RO$_2$ fate in
OFRs compared to the atmosphere. The only exception in OFRs occurs at very high VOC precursor
concentrations (OHR$_{ext}$ significantly >100 s$^{-1}$) in OFR254 (Fig. S2), where OH levels are not substantially
suppressed due to large amounts of O$_3$ (Peng et al., 2015). As a result, RO$_2$ concentration is remarkably
increased by strong production and RO$_2$+RO$_2$ relative importance increases roughly quadratically and
becomes significant.

The generally lower relative importance of RO$_2$+RO$_2$ in OFRs than in the atmosphere is more

obvious for the fate of RO$_2$ with fast RO$_2$+RO$_2$ rate constants (Figs. 1b,d and 3). Although OFRs can
reasonably reproduce RO$_2$ fates in low-VOC ambient environments (e.g. typical pristine and forested
areas; Figs. 1b,d and 3) and low-OHR$_{ext}$ chambers, OFR185 cannot achieve relative importance of
RO$_2$+RO$_2$ significantly larger than 50%, corresponding to higher-VOC environments (e.g. P$_1$ in Fig. 1) and



high-OHR$_{ext}$ chamber experiments (e.g. $C_2$ and $C_5$ in Fig. 1; the distribution for $C_2$ is also shown in Fig. 3).
In OFR254-70, a relative importance of RO$_2$+RO$_2$ as high as ~90% may be attained (Fig. S3). However,
this requires very high OHR$_{ext}$, which leads to medium (and slower) RO$_2$+RO$_2$ showing higher-than-
ambient relative importance. In reality, fast RO$_2$+RO$_2$ all involve substituted RO$_2$, which almost certainly
arise from and coexist with unsubstituted RO$_2$ (with slower self-/cross reactions). Therefore, very high
OHR$_{ext}$ in OFR254 is not really suitable for attaining dominant RO$_2$+RO$_2$ conditions. In OFR185, a higher
OHR$_{ext}$ generally also results in a higher RO$_2$+RO$_2$ relative importance because of higher RO$_2$ production
(Fig. S3). Nevertheless, higher OHR$_{ext}$ is more likely to lead to risky or bad conditions (Fig. 3; Peng et al.,
2016). It should be noted that although it is difficult to reliably achieve RO$_2$+RO$_2$ with a relative
importance larger than 50% in RO$_2$ fate in OFRs, the distributions of RO$_2$+RO$_2$ relative importance in
OFRs seems to be within a factor of 2 of those of field/aircraft campaigns (Fig. 3).
In the case of very fast RO$_2$+RO$_2$, all features for fast RO$_2$+RO$_2$ discussed above are still present
(Fig. S1c,d). The only major difference between the results for fast RO$_2$+RO$_2$ and very fast RO$_2$+RO$_2$ is
the significantly higher relative importance of RO$_2$+RO$_2$ in RO$_2$ fate in the latter case, which is expected.
In summary, the fast RO$_2$+RO$_2$ is not perfectly reproduced in OFRs in terms of relative importance in RO$_2$
fate, but it is significant when this pathway is also important in the atmosphere.
The HO$_x$ recycling ratio β (see Sect. 2.3) is one of the key factors determining HO$_2$ in the OFR
model, yet it is not well constrained. Although we make reasonable assumptions for it in the model
input (see Section 2.3 for details), a sensitivity study to explore its effects is also performed here. For
RO$_2$ with the fast self-/cross-reaction rate constant, we perform the simulations with the HO$_x$ recycling
ratios fixed to a number of values from 0 (radical termination) to 2 (radical proliferation) in lieu of those
calculated under the assumptions described in Section 2.3. As expected, the contribution of RO$_2$+RO$_2$
to RO$_2$ fate increases monotonically between β=2 and β=0 (Fig. S4), as the recycling of the competing
reactant HO$_2$ decreases. Nevertheless, the change in the average RO$_2$+RO$_2$ relative importance from β=0
to β=2 is generally within a factor of 2. Thus, it still holds that the RO$_2$+RO$_2$ relative importance in OFRs
is generally lower than in the atmosphere. Only at β~0 may OFR185 theoretically attain a relative
importance of RO$_2$+RO$_2$ of ~70%, as in the $P_1$ case (pristine, but relatively high-VOC, Figure S5). Note
that β=0 for all VOC oxidation (including oxidation of intermediates) is extremely unlikely. In OFR254,
even if RO$_2$+RO$_2$ may contribute up to ~100% to RO$_2$ fate at very high OHR$_{ext}$ at β=0, these conditions
still also lead to significant RO$_2$+RO$_2$ in the fate of RO$_2$ that self-/cross-react more slowly, which is not
atmospherically relevant.

**338    3.1.2   Acyl RO$_2$**

As described in Section 2.1, the generic acyl RO$_2$ modeled in this study has the same loss
pathways as RO$_2$ with the fast self-/cross-reaction rate constant, except for RO$_2$+NO$_2$, which can be a
significant acyl RO$_2$ loss pathway in OFRs as well as both chambers and atmosphere. When this reaction
is included in the simulations of acyl RO$_2$, it is a minor or negligible loss pathway of RO$_2$ at low N$_2$O,
while it can be the dominant fate of acyl RO$_2$ at high N$_2$O (Fig. 4). In general, the RO$_2$+NO$_2$ relative
importance increases with initial N$_2$O. This is always true in OFR254-70-iN$_2$O between N$_2$O=0.02% and



$N_2O=20\%$, while in OFR185-i$N_2$O, the average relative contribution of $RO_2+NO_2$ to $RO_2$ fate starts to
decrease at $N_2O\sim10\%$, because $RO_2+NO$ regains some importance. This results from the $HO_x$
suppression caused by high $NO_y$ and strong NO production at high $N_2O$. Strong NO production increases
its concentration and suppresses $HO_x$ under these conditions, limiting the conversion of NO to $NO_2$.
Because of the strong OH suppression by high $NO_y$ at $N_2O\geq10\%$, these conditions are not desirable (Peng
et al., 2018).

The only difference between the simulations of acyl $RO_2$ and of the fast-self-/cross-reacting non-

acyl $RO_2$ is the quasi-irreversible reaction $RO_2+NO_2\rightarrow RO_2NO_2$, whose effects are revealed by a
comparison of the triangle plots of the $RO_2$ fates in each case (Figs. 1b,d and S6). $RO_2+NO_2$ is clearly
dominant in acyl $RO_2$ fate in OFRs as long as $RO_2+NO$ plays some role (not necessarily under high-NO
conditions). In OFR185-i$N_2$O, the relative importance of $RO_2+RO_2$ in the sum of the $HO_2$, NO and $RO_2$
pathways is reduced (Fig. S6a), compared to that of non-acyl $RO_2$ with the fast $RO_2+RO_2$ (Fig. 1b),
because $RO_2+NO_2$ decrease acyl $RO_2$ concentration. Such a decrease is not significant in OFR254-70-
i$N_2$O (Fig. S6b, compared to Fig. 1d), since for non-acyl $RO_2$, it is already stored in the form of $RO_2NO_2$
as $RO_2$ reservoir. In other words, the high initial $O_3$ greatly accelerates NO-to-$NO_2$ oxidation, and shifts
the equilibrium $RO_2+NO_2\leftrightarrow RO_2NO_2$ far to the right even for non-acyl $RO_2$.

$RO_2+NO_2$ is an inevitable sink of most acyl $RO_2$ in high-$NO_x$ OFRs. Its contribution to acyl $RO_2$ fate

in OFRs is often higher than in urban atmospheres, where the relative amounts of NO and $NO_2$ vary
overtime. At midday, most NO is usually oxidized to $NO_2$ in urban atmospheres and $RO_2+NO_2$ dominates
acyl $RO_2$ fate, as in high-$NO_x$ OFRs. During morning rush hours and/or near major NO sources, NO may
be significantly more abundant than $NO_2$ and $RO_2+NO$ is likely the dominant acyl $RO_2$ loss pathway,
which cannot be simulated in OFRs with the current range of techniques.

Acyl $RO_2$ are not the dominant type among $RO_2$s under most conditions in OFRs, chambers and

the atmosphere, since their formation usually requires multistep (at least 2 steps) oxidation via specific
pathways leading to an oxidized end group (i.e. aldehyde and then acylperoxy). However, acyl $RO_2$ can
still be a major (very roughly 1/3) source of $RO_2$ at ages of several hours or higher according to
estimations made using the GECKO-A model in urban and forested atmospheres. Therefore, acyl $RO_2$
chemistry in high-NO OFR can significantly deviate from that in an urban atmosphere with NO
dominating $NO_x$, and can be relevant to an urban atmosphere with $NO_2$ dominating $NO_x$. On the other
hand, a few theoretical studies suggested that H-abstraction by the acylperoxy radical site from
hydroperoxy groups close to the acylperoxy site in multifunctional acyl $RO_2$ may be extremely fast
(Jørgensen et al., 2016; Knap and Jørgensen, 2017). If these theoretical predictions are sufficiently
accurate, these acyl $RO_2$ may exclusively undergo intramolecular H-shift to form non-acyl $RO_2$ or other
radicals and prevent $RO_2+NO_2$ from occurring even at very high (ppm-level) $NO_2$. However, this type of
$RO_2$ is structurally specific and may not have strong impacts on the overall acyl $RO_2$ chemistry.
**3.2    Simulations with all significant pathways**

Since $RO_2$ isomerization does not significantly affect the generic $RO_2$ concentration, the two $RO_2$

fates that were recently found to be potentially important, i.e. $RO_2+OH$ and $RO_2$ isomerization, can be





discussed separately.

### 3.2.1    $RO_2$+OH

In the troposphere, $RO_2$+OH is a minor (at low NO) or negligible (at high NO) $RO_2$ loss pathway

(Fittschen et al., 2014; Assaf et al., 2016; Müller et al., 2016), as its rate constant is roughly an-order-of-
magnitude higher than that of $RO_2$+$HO_2$ (Table 1) while ambient OH concentration is on average 2-
orders-of-magnitude lower than that of $HO_2$ (Mao et al., 2009; Stone et al., 2012; Fig. 5). We will not
discuss $RO_2$+OH in the high-NO cases in detail. Simply put, the relative importance of $RO_2$+OH is
generally negatively correlated with input $N_2O$ in OFR-i$N_2O$, as $NO_x$ suppresses OH and the relative
importance of $RO_2$+NO increases. Below, we focus on low-NO (actually, for simplicity, zero-NO)
conditions.

At $N_2O$=0, it would be ideal if an $HO_2$-to-OH ratio identical to the ambient values was realized in

OFRs. In OFR185 cases with medium $RO_2$+$RO_2$, $HO_2$-to-OH ratio around 100 occurs at a combination of
low $H_2O$ (on the order of 0.1%), low F185 (on the order of $10^{11}$ photons $cm^{-2}$ $s^{-1}$), and medium $OHR_{ext}$
(10–100 $s^{-1}$); and also at medium F185 (~$10^{12}$ photons $cm^{-2}$ $s^{-1}$) combined with very high $OHR_{ext}$ (~1000
$s^{-1}$, Fig. S7). Under both sets of conditions, relatively high external OH reactants suppress OH, whose
production is relatively weak, and convert some OH into $HO_2$ through $HO_x$ recycling in organic oxidation
(e.g. via alkoxy radical chemistry). The reason why such an OH-to-$HO_2$ conversion is needed to attain an
ambient-like $HO_2$-to-OH ratio is that OFR185 is unable to achieve this via the internal (mainly assisted
by $O_3$) interconversion of $HO_x$. This inability is most evident when F185 ($10^{13}$–$10^{14}$ photons $cm^{-2}$ $s^{-1}$) and
$H_2O$ (on the order of 1%) are high and $OHR_{ext}$ is low (<~10 $s^{-1}$; Fig. S7). Under these conditions, OH
production by $H_2O$ photolysis is so strong that the $HO_2$-to-OH ratio is lowered to ~1, since OH and H
(which recombines with $O_2$ to form $HO_2$) are produced in equal amounts from $H_2O$ photolysis. As the
$RO_2$+OH rate constant is only roughly 1-order-of-magnitude higher than that for $RO_2$+$HO_2$, slightly lower
$HO_2$-to-OH ratios (e.g. ~30) suffice to keep $RO_2$+OH minor in this case. A combination of UV and $H_2O$
that are not very high and a moderate $OHR_{ext}$ that is able to convert some OH to $HO_2$ and somewhat
elevate the $HO_2$-to-OH ratio results in minor relative importance $RO_2$+OH (Figs. S7 and S8).

In OFR254-70, it is more difficult to reach an $HO_2$-to-OH ratio of ~100, which can only be realized

at a combination of very low $H_2O$ and F254 (~0.07% and ~$5\times10^{13}$ photons $cm^{-2}$ $s^{-1}$, respectively) and very
high $OHR_{ext}$ (~1000 $s^{-1}$). This is mainly due to high $O_3$ in OFR254-70, which controls the $HO_x$
interconversion through $HO_2$+$O_3$→OH+$2O_2$ and OH+$O_3$→$HO_2$+$O_2$ and makes both OH and $HO_2$ more
resilient to changes due to $OHR_{ext}$ (Peng et al., 2015). Even without $H_2O$ photolysis at 185 nm as a major
$HO_2$ source, the $HO_x$ interconversion controlled by $O_3$ in OFR254-70 still brings $HO_2$-to-OH ratio to ~1 in
the case of minimal external perturbation (see the region at the highest $H_2O$ and UV and $OHR_{ext}$=0 in
the OFR254-70 part of Fig. S7). This ratio cannot be easily elevated in OFR254-70 because of the
resilience of OH to suppression for this mode (Peng et al., 2015). Thus, this ratio is relatively low (<30)
under most conditions (Fig. S7), and consequently (and undesirably), $RO_2$+OH is a major $RO_2$ fate in
OFR254-70. There is an exception at relatively low $H_2O$ and UV with very high $OHR_{ext}$ (Fig. S8), however
these conditions are undesirable in terms of non-tropospheric organic photolysis (Peng et al., 2016).





421   Only the results of $RO_2$ with the medium $RO_2+RO_2$ are discussed in this subsection. Those of RO2

422 with the fast $RO_2+RO_2$ are not shown as they are not qualitatively different. In OFR185, for the fast-self-

423 /cross-reacting $RO_2$, $RO_2+RO_2$ is relatively important at high $OHR_{ext}$ (>~100 s$^{-1}$; Fig. S3), while $RO_2+OH$ is

424 a major $RO_2$ fate at low $OHR_{ext}$ (generally on the order of 10 s$^{-1}$ or lower) and relatively high $H_2O$ and UV

425 (Fig. S8). These two ranges of conditions are relatively far away from each other, and hence there is no

426 condition under which $RO_2+RO_2$ and $RO_2+OH$ are both major pathways that compete, which simplifies

427 understanding $RO_2$ fate. However, in OFR254-70, some conditions may lead to both significant $RO_2+RO_2$

428 (for the fast-self-/cross-reacting $RO_2$) and $RO_2+OH$ (e.g. $H_2O$~0.5%, F254~1x10$^{15}$ photons cm$^{-2}$ s$^{-1}$ and

429 $OHR_{ext}$~100 s$^{-1}$). Nevertheless, as long as $RO_2+OH$ plays a major role, these conditions do not bear much

430 experimental interest and thus do not need to be discussed in detail.

431 **3.2.2 $RO_2$ isomerization**

432   $RO_2$ isomerization is a first-order reaction. For this type of reactions to occur, $RO_2$ does not need

433 any other species but only a sufficiently long lifetime against all other reactants combined, as most $RO_2$

434 isomerization rate constants are <10 s$^{-1}$. Radical (OH, $HO_2$, NO etc.) concentrations in OFRs are much

435 higher than ambient levels and may shorten $RO_2$ lifetimes compared to those in the troposphere.

436 Possibly reduced $RO_2$ lifetimes naturally raise concerns over the potentially diminished importance of

437 $RO_2$ isomerization in OFRs.

438   In this section we examine generic $RO_2$ lifetimes against all reactions (calculated without $RO_2$

439 isomerization taken into account) in OFR (including OFR-i$N_2O$) cases (for the medium $RO_2+RO_2$ case) and

440 compare them with the $RO_2$ lifetimes in recent major field/aircraft campaigns in relatively clean

441 environments and a field campaign in an urban area (CalNex-LA), as well as a low-NO chamber

442 experiment (Fig. 6). Indeed, $RO_2$ lifetime in clean ambient cases and in chambers with near-ambient

443 radical levels are generally much longer than those in OFRs. The $RO_2$ lifetime distribution of the explored

444 good and risky cases in OFR254-70 (including OFR254-70-i$N_2O$) barely overlaps with the ambient and

445 chamber cases, while in OFR185 (including OFR185-i$N_2O$), $RO_2$ lifetime can be as long as ~10 s, which is

446 longer than in urban areas and roughly at the lower end of the range of ambient $RO_2$ lifetime in clean

447 environments (Fig. 6). The longest $RO_2$ lifetime in OFR185 occurs at very low F185 (on the order of 10$^{11}$

448 photons cm$^{-2}$ s$^{-1}$) and $H_2O$ (~0.1%; Fig. S9), where $HO_x$ is low. In OFR254-70, for $RO_2$ to survive for ~10 s,

449 in addition to very low UV and $H_2O$, high $OHR_{ext}$ is also needed (Fig. S9). High-$OHR_{ext}$ conditions in

450 OFR254-70 cause OH suppression and a decrease in $HO_x$ concentration, and hence result in relatively

451 long $RO_2$ lifetimes. However, the strong OH suppression is likely to give bad conditions (high contribution

452 of non-tropospheric photolysis).(Peng et al., 2016) Low-$OHR_{ext}$ conditions do not lead to long $RO_2$

453 lifetimes in OFR254-70 even at very low F254 and $H_2O$, since $O_3$-assisted $HO_x$ recycling prevents a very

454 low $HO_x$ level even if $HO_x$ primary production is low.(Peng et al., 2015)

455   An $RO_2$ lifetime (without $RO_2$ isomerization included) of 10 s leads to a relative importance of

456 isomerization of 50% in the total fate (including all loss pathways) of $RO_2$ with an isomerization rate

457 constant of 0.1 s$^{-1}$, which is a typical order of magnitude for isomerization rate constants of

458 multifunctional $RO_2$ with hydroxyl and hydroperoxy substituents (Fig. 6; Crounse et al., 2013; D'Ambro



et al., 2017; Praske et al., 2018). Although a 50% relative importance of isomerization under some OFR
conditions is still lower than those in relatively low-NO ambient environments and low-NO chambers,
this relative importance should certainly be deemed major and far from negligible as some have
speculated (Crounse et al., 2013). Other monofunctional $RO_2$ (with peroxy radical site only) and
bifunctional $RO_2$ with peroxy radical site and a carbonyl group isomerize so slowly ($\sim$0.001–0.01 s$^{-1}$) that
their isomerizations are minor or negligible loss pathways in the atmosphere, chambers and OFRs with
$RO_2$ lifetimes around 10 s (Fig. 6). Isomerizations of other types of multifunctional $RO_2$ (e.g.
multifunctional acyl $RO_2$ with hydroxyl and hydroperoxy substituents at favorable positions) are
extremely fast (rate constants up to $10^6$ s$^{-1}$; Jørgensen et al., 2016; Knap and Jørgensen, 2017) and
always dominate in their fates in the relatively low-NO atmosphere and chambers and OFRs with $RO_2$
lifetimes around 10 s.

In the discussion about $RO_2$ isomerization above (as in the RO2+OH exploration in Section 3.2.1),

we only examine low-NO (or zero-NO for simplicity) conditions with medium $RO_2$+$RO_2$. In high-NO
environments, e.g. polluted urban atmospheres with NO of at least $\sim$10 ppb and high-NO OFRs in the
i$N_2$O modes, $RO_2$ lifetime is so short that isomerization is no longer a major fate for any but the most
rapidly isomerizing multifunctional $RO_2$ discussed above. NO measured in Los Angeles during the
CalNex-LA campaign (Ortega et al., 2016) was only $\sim$1 ppb, which would to allow $RO_2$ to survive for a
few seconds and isomerize (Fig. 6), even in an urban area.

The OFR simulations for the discussions about $RO_2$ isomerization are the same as those

conducted to study $RO_2$+OH, i.e. the ones with the medium $RO_2$+$RO_2$ and $RO_2$+OH included. For fast $RO_2$
self-/cross-reaction cases, $RO_2$ lifetimes may be significantly shorter than for $RO_2$ with the medium self-
/cross-reaction rate constant at high OHR$_{ext}$ ($>\sim$100 s$^{-1}$) in OFR185 (Fig. S3). These high-OHR$_{ext}$ conditions
are likely to be risky or bad (of little experimental interest)(Peng et al., 2016) and thus do not need to
be discussed further in detail. OFR254-70 (a zero-NO mode) does not generate good or risky (of at least
some experimental interest in terms of non-tropospheric organic photolysis) conditions also leading to
low-NO-atmosphere-relevant $RO_2$ lifetimes (Fig. 6). $RO_2$ with faster self-/cross-reaction rate constants
have even shorter lifetimes in OFR254-70 and will not be discussed further.
**3.3    Guidelines for OFR operation**

In this subsection we discuss OFR operation guidelines for atmospherically relevant $RO_2$ chemistry,

with a focus on OFR185 and OFR254 (zero-NO modes). Since $RO_2$+$HO_2$ and $RO_2$+NO both can vary from
negligible to dominant $RO_2$ fate in OFRs, chambers and the atmosphere (Figs. 1 and 2), these two
pathways are not a concern in OFR atmospheric relevance considerations. Neither is the $RO_2$+$RO_2$ a
major concern. Medium or slower $RO_2$+$RO_2$ is minor or negligible in the atmosphere and chambers, as
well as in OFRs, as long as high OHR$_{ext}$ is avoided in OFR254 (Fig. S2). Fast $RO_2$+$RO_2$ is somewhat less
important in OFRs than in the atmosphere (Figs. 1b,d and 3), but is still qualitatively atmospherically
relevant, given the uncertainties associated with the $HO_x$ recycling ratios of various reactive systems
and the huge variety of $RO_2$ types (and hence $RO_2$+$RO_2$ rate constants).

Accordingly, we focus on the atmospheric relevance of $RO_2$+OH and $RO_2$ isomerization, i.e. their



relative contributions close to ambient values. Under typical high-NO conditions, $RO_2$+NO dominates
$RO_2$ fate and $RO_2$+OH is negligible. High NO also shortens $RO_2$ lifetime enough to effectively inhibit $RO_2$
isomerization. Both the dominance of $RO_2$+NO and the inhibition of $RO_2$ isomerization also occur in the
atmosphere and in chambers, so high-NO OFR operation (typically NO>10 ppb) represents these
pathways realistically. Some care is, however, required with the $RO_2$+OH and $RO_2$ isomerization
pathways at low NO. Since $RO_2$+$HO_2$ in OFRs is always a major $RO_2$ fate at low NO and $RO_2$+$RO_2$ are
generally not problematic, $RO_2$+OH and $RO_2$+$HO_2$ can be kept atmospherically relevant as long as $HO_2$-
to-OH ratio is close to 100 (the ambient average). In addition, $RO_2$ lifetime (calculated without $RO_2$
isomerization taken into account) should be at least around 10 s.

Practically, OH production should be limited to achieve this goal. Too strong OH production at high

$H_2O$ and UV can elevate OH and $HO_2$ concentrations, which shortens $RO_2$ lifetime, and decreases the
$HO_2$-to-OH ratio to ~1 (see Sect. 3.2.1). OH production is roughly proportional to both $H_2O$ and UV (Peng
et al., 2015), so can be limited by reducing either or both. However, $H_2O$ and UV have different effects
on non-tropospheric organic photolysis. At a certain $OHR_{ext}$, OH production rate roughly determines OH
concentration in OFRs. Reducing UV decreases both OH and UV roughly proportionally (Peng et al.,
2015), and hence changes in $F185_{exp}/OH_{exp}$ and $F254_{exp}/OH_{exp}$ are small (Peng et al., 2016); i.e. non-
tropospheric organic photolysis does not become significantly worse if UV is reduced. By contrast, if $H_2O$
is reduced without also decreasing UV, $F185_{exp}/OH_{exp}$ and $F254_{exp}/OH_{exp}$ both increase, signifying
stronger relative importance of non-tropospheric photolysis. Therefore, reducing UV is strongly
preferred as an OH production limitation method, and is effective in making both $RO_2$+OH and $RO_2$
isomerization more atmospherically relevant.

To further explore the effects of UV reduction on the $RO_2$+OH (Fig. 5) and $RO_2$ isomerization (Fig.

6) pathways, we divide our OFR case distributions into higher-UV and lower-UV classes, with the
boundary being the mid-level (in logarithmic scale) UV in the explored range. The distributions for
lower-UV conditions (solid lines in Figs. 5 and 6) are clearly closer to the ambient cases (i.e. $HO_2$-to-OH
ratio closer to 100, smaller $RO_2$+OH relative importance and longer $RO_2$ lifetime).

Since OFR254 is unable to achieve conditions with both at least some experimental interest (i.e.

with sufficiently low non-tropospheric photolysis) and atmospherically relevant $RO_2$ lifetime, we now
discuss preferable conditions for OFR185 only. As F185 close to or lower than $10^{12}$ photons cm$^{-2}$ s$^{-1}$ is
needed for $RO_2$ lifetime to be around 10 s or longer (Fig. S9), the OH concentration under preferable
conditions for atmospherically relevant $RO_2$ chemistry (~$10^9$ molecules cm$^{-3}$ or lower) is much lower
than the maximum that OFR185 can physically reach (~$10^{10}$–$10^{11}$ molecules cm$^{-3}$). Furthermore, lower
OH production leads to higher susceptibility to OH suppression by external OH reactants (Peng et al.,
2015), which can create non-tropospheric photolysis problems (Peng et al., 2016). We thus recommend
as high $H_2O$ as possible to maintain practically high OH while allowing lower UV to limit the importance
of non-tropospheric organic photolysis.

The performance of various OFR185 conditions at high $H_2O$ (2.3%) is illustrated in Fig. 7 as a

function of F185 and $OHR_{ext}$. The three criteria for the performance, i.e. $RO_2$ lifetime (calculated without





RO$_2$ isomerization considered), relative importance of RO$_2$+OH and log(F254$_{exp}$/OH$_{exp}$) (a measure of
254 nm non-tropospheric photolysis, which is usually worse than that at 185 nm; Peng et al., 2016) are
shown. At F185 of ~10$^{11}$–10$^{12}$ photons cm$^{-2}$ s$^{-1}$ and OHR$_{ext}$ around or lower than 10 s$^{-1}$, all three criteria
are satisfied. Since UV (and hence OH production) is relatively low, a low OHR$_{ext}$ (~10 s$^{-1}$) is required to
avoid heavy OH suppression and keep conditions good (green area in the bottom panel of Fig. 7).
Nevertheless, risky conditions [log(F254$_{exp}$/OH$_{exp}$)<7; light red area in the bottom panel of Fig. 7] may
also bear some experimental conditions depending on the type of VOC precursors (specifically on their
reactivity toward OH and their photolability at 185 and 254 nm, and the same quantities for their
oxidation intermediates; Peng et al., 2016; Peng and Jimenez, 2017). Thus, higher OHR$_{ext}$ (up to ~100 s$^{-1}$)
may also be considered in OFR experiments with some precursors (e.g. alkanes). In practice, the
preferred conditions may require F185 even lower than that our lowest simulated lamp setting (Li et al.,
2015). Such a low F185 may be realized e.g. by partially blocking 185 nm photons using non-transparent
lamp sleeves with evenly placed holes that allow some 185 nm transmission.

Under these preferred conditions, OH concentration in OFR185 is ~10$^9$ molecules cm$^{-3}$, equivalent

to a photochemical age of ~1 eq. d for a typical residence time of 180 s. This is much shorter than ages
corresponding to the maximal oxidation capacity of OFRs (usually eq. weeks or months; Peng et al.,
2015) but it is similar to the ages of the maximal organic aerosol formation in OFRs processing ambient
air (Tkacik et al., 2014; Ortega et al., 2016; Palm et al., 2016). We show the maximal SOA formation case
in the OFR185 experiments in the BEACHON-RoMBAS campaign in the Rocky Mountains (Palm et al.,
2016) as an example (Figs. 5 and 6). During the campaign, relative humidity was high (>60% in most of
the period), OHR$_{ext}$ was estimated to be relatively low (~15 s$^{-1}$) in this forested area, and UV in the OFR
was limited in the case of the maximal SOA formation age (~0.7 eq. d). All these physical conditions
were favorable for atmospherically relevant RO$_2$ fate (Figs. 5 and 6). RO$_2$+OH was minor in this case and
the relative importance of RO$_2$ isomerization in RO$_2$ fate in the OFR was within a factor of ~2 of that in
the atmosphere for all RO$_2$ (regardless of isomerization rate constant) during the BEACHON-RoMBAS
campaign (Fig. 6). The effect of UV on the relative importance of RO$_2$ isomerization for this example is
also illustrated in Fig. 6. In the sensitivity case with a lower age, a lower UV results in a larger
contribution of isomerization to RO$_2$ fate, while the relative importance of RO$_2$ isomerization is lower in
a sensitivity case with an age 3 times of that of the maximal SOA formation. In an extreme sensitivity
case with the highest UV in the range of this study (with an age of 4 eq. mo), RO$_2$ isomerization becomes
minor or negligible for all RO$_2$ except extremely rapidly isomerizing ones.

The discussions above indicate that the atmospheric relevance of gas-phase RO$_2$ chemistry in OFRs

deteriorates as the photochemical age over the whole residence time (180 s) increases. To reach longer
ages, longer residence times (with UV being still low) can be adopted. However, OFR residence times >
10 min tend to be limited by the increasing importance of wall losses (Palm et al., 2016). As a result,
longer residence times can only increase photochemical age in OFRs up to about a week. This implies
that in OFR cases with ages much higher than that of maximal SOA formation (corresponding to the
heterogeneous oxidation stage of SOA), the atmospheric relevance of gas-phase RO$_2$ chemistry in the





SOA formation stage (before the age of maximal SOA formation) often cannot be ensured. However,
under those conditions typically new SOA formation is not observed, and the dominant process
affecting OA is heterogeneous oxidation of the pre-existing OA (Palm et al., 2016). If the heterogeneous
oxidation of the newly formed SOA is of interest, a two-stage solution may be required. Lower UV can
be used in the SOA formation stage to keep the atmospheric relevance of the gas-phase chemistry, while
high UV can be used in the heterogeneous aging stage to reach a high equivalent age. The latter
approach is viable since heterogeneous oxidation of SOA by OH is slow and particle-phase chemistry is
not strongly affected by gas-phase species except OH, when OH is very high (Richards-Henderson et al.,
2015, 2016; Hu et al., 2016). This two-stage solution may be realized through a cascade-OFR system or
UV sources at different intensities within an OFR (e.g. spliced lamps).

Praske et al. (2018) measured $RO_2$ isomerization rate constants at 296 and 318 K and observed an

increase in the rate constants by a factor of ~5 on average. A 15 K temperature increase in OFRs would
lead to $RO_2$ isomerization being accelerated by a factor of ~3, while other major gas-phase radical
reactions have weak or no temperature-dependence. As a consequence, the relative importance of $RO_2$
isomerization in $RO_2$ fate in OFRs can be elevated and closer to atmospheric values (Fig. 6). Nevertheless,
a 15 K increase in temperature may also result in some OA evaporation (Nault et al., 2018).

As discussed above, high $H_2O$, low UV and low $OHR_{ext}$ are recommended for keeping the

atmospheric relevance of $RO_2$ chemistry in OFRs. These three requirements are also part of the
requirements for attaining good high-NO conditions in OFR185-iNO (the OFR185 mode with initial NO
injection; Peng and Jimenez, 2017). In addition to these three, an initial NO of several tens of ppb is also
needed to obtain a good high-NO condition in OFR185-iNO. Under these conditions, $RO_2$+NO dominates
over $RO_2$+$HO_2$, and hence $RO_2$+OH; UV is low, the photochemical age is typically ~1 eq. d, and $RO_2$
lifetime can be a few seconds. Therefore, these conditions are a good fit for studying the environments
in relatively clean urban areas, such as Los Angeles during CalNex-LA (Ortega et al., 2016), where NO is
high enough that the dominant bimolecular fate of $RO_2$ is $RO_2$+NO but low enough to maintain $RO_2$
lifetimes that allow most common $RO_2$ isomerizations.

As $RO_2$ fate in OFRs is a highly complex problem and it can be tricky to find suitable physical

conditions to simultaneously achieve experimental goals and keep the atmospheric relevance of the
chemistry in OFRs, we provide here an OFR $RO_2$ Fate Estimator (in Supplement) to qualitatively aid
experimental planning. The OFR $RO_2$ Fate Estimator couples the OFR Exposure Estimator(Peng et al.,
2016, 2018) to a General $RO_2$ Fate Estimator (also in Supplement, see Fig. S10 for a screenshot of its
layout). The OFR Exposure Estimator updated in this study also contains estimation equations for the
$HO_2$-to-OH ratio in OFR185 (in OFR254, $RO_2$ fate is always atmospherically irrelevant at low NO, while
at high NO, $RO_2$+NO dominates and a detailed $RO_2$ fate analysis is no longer needed). In the General $RO_2$
Fate Estimator, all $RO_2$ reactant concentrations and all $RO_2$ loss pathway rate constants can be specified.
Thus the General $RO_2$ Fate Estimator can also be applied to the atmosphere and chamber experiments,
in addition to OFRs. When applied to OFRs, the General $RO_2$ Fate Estimator is provided by the OFR $RO_2$
Fate Estimator with quantities estimated in the OFR Exposure Estimator (e.g. OH and NO). $RO_2$





concentration and fate are calculated according to Appendix A in the $RO_2$ Fate Estimators.

**4    Conclusions**

We investigated $RO_2$ chemistry in OFRs with an emphasis on its atmospheric relevance. All
potentially major loss pathways of $RO_2$, i.e. reactions of $RO_2$ with $HO_2$, NO and OH, that of acyl $RO_2$ with
$NO_2$, self-/cross-reactions of $RO_2$ and $RO_2$ isomerization, were studied and their relative importance in
$RO_2$ fate were compared to those in the atmosphere and chamber experiments. OFRs were shown to
be able to tune the relative importance of $RO_2+HO_2$ vs. $RO_2+NO$ by injecting different amounts of $N_2O$.
For many $RO_2$ (including all unsubstituted non-acyl $RO_2$ and substituted secondary and tertiary $RO_2$),
their self-reactions and the cross-reaction between them are minor or negligible in the atmosphere and
chambers. This is also the case in OFR185 (including OFR185-i$N_2O$) and OFR254-i$N_2O$, however those
$RO_2$ self-/cross-reactions can be important at high precursor concentrations ($OHR_{ext}>100$ s$^{-1}$) in OFR254.
For substituted primary $RO_2$ and acyl $RO_2$, their self-/cross-reactions (including the ones with $RO_2$ whose
self-reaction rate constants are slower) can play an important role in $RO_2$ fate in the atmosphere and
chambers, and may also be major $RO_2$ loss pathways in OFRs, although they are somewhat less
important in OFRs than in the atmosphere. Acylperoxy nitrates are the dominant sink of acyl $RO_2$ at high
$NO_x$ in OFRs, while only a minor reservoir of acyl $RO_2$ in the atmosphere under most conditions except
in urban atmospheres, where acylperoxy nitrate formation can be the dominant acylperoxy loss
pathway when most NO is oxidized to $NO_2$. In chambers, most acyl $RO_2$ can be stored in the form of
acylperoxy nitrates if $NO_2$ is very high (hundreds of ppb to ppm level).
Under typical high-NO conditions, $RO_2+NO$ dominates $RO_2$ fate and $RO_2$ lifetime is too short to
allow most $RO_2$ isomerizations, regardless of whether in the atmosphere, chambers or OFRs, thus raising
no concern over the atmospheric relevance of the OFR $RO_2$ chemistry. However, under low-NO
conditions, OFR254 cannot yield any physical conditions leading to sufficiently long $RO_2$ lifetime for its
isomerization because of the high radical levels and their resilience to external perturbations in OFR254.
In OFR185 with strong OH production (and hence high OH), $RO_2+OH$ and $RO_2$ isomerization may strongly
deviate from the atmosphere (becoming important and negligible, respectively, for relatively rapidly
isomerizing $RO_2$). To attain both atmospherically relevant VOC and $RO_2$ chemistries, OFR185 requires
high $H_2O$, low UV and low $OHR_{ext}$, which conditions ensure minor or negligible $RO_2+OH$ and a relative
importance of $RO_2$ isomerization in $RO_2$ fate in OFRs within x~2 of that in the atmosphere but limit the
maximal photochemical age that can be reached to a few eq. days. This age roughly covers SOA
formation in ambient air up to its maximum. To reach a much higher age for studying SOA
functionalization/fragmentation by heterogeneous oxidation, a sequence of low-UV SOA formation
followed by a high UV condition (in the same reactor or in cascade reactors) would be needed. High
$H_2O$, low UV and low $OHR_{ext}$ in the OFR185-iNO mode can achieve conditions relevant to clean urban
atmosphere, i.e. high-NO but not sufficiently high to inhibit common $RO_2$ isomerization.
Finally, $RO_2$ chemistry is not only highly complex but also plays a central and instrumental role in
atmospheric chemistry, in particular VOC oxidation and SOA formation. For all experiments conducted
with atmospheric chemistry simulation apparatus (chambers, flow reactors etc.), an atmospherically



relevant RO$_2$ chemistry is crucial to meaningful experimental results. However, most literature studies
did not publish experimental data that are sufficient for estimating RO$_2$ fate. The FIXCIT chamber
experiment campaign is one of the few exceptions where comprehensive data were reported (Nguyen
et al., 2014) and used for the RO$_2$ fate analysis in the present work. We recommend measuring and/or
estimating and reporting OH, HO$_2$, NO, NO$_2$ and OHR$_{VOC}$ (or initial precursor composition at least)
whenever possible, for all future atmospheric laboratory and field experiments for organic oxidation to
facilitate the analysis of RO$_2$ fate and the evaluation of its atmospheric relevance.

**Appendix A: Steady-state approximation for generic RO$_2$**
The production rate of a generic RO$_2$ is almost identical to the VOC consumption rate, since the
second step of the conversion chain VOC→R→RO$_2$ is extremely fast. Therefore, the generic RO$_2$
production rate, $P$, can be expressed as follows:
$$P = \sum_i k_i c_i \cdot \mathrm{OH} = \mathrm{OHR_{VOC}} \cdot \mathrm{OH} \tag{A1}$$
where OH is OH concentration and $c_i$ and $k_i$ are respectively the concentration and the reaction rate
constant with OH of the $i^{th}$ VOC. OHR$_{VOC}$ is the total OHR due to VOC and equal to $\Sigma_i k_i c_i$ by definition.
For the generic RO$_2$ loss rate, the reactions of RO$_2$ with HO$_2$, NO, RO$_2$, NO$_2$ (for acyl RO$_2$ only) and
OH are considered. Isomerization generally does not lead to a total RO$_2$ concentration decrease and is
thus not included in its loss rate. Then the RO$_2$ loss rate is
$$L = k_{\mathrm{HO_2}}\mathrm{RO_2} \cdot \mathrm{HO_2} + k_{\mathrm{NO}}\mathrm{RO_2} \cdot \mathrm{NO} + 2k_{\mathrm{RO_2}}\mathrm{RO_2} \cdot \mathrm{RO_2} + k_{\mathrm{NO_2}}\mathrm{RO_2} \cdot \mathrm{NO_2} + k_{\mathrm{OH}}\mathrm{RO_2} \cdot \mathrm{OH} \tag{A2}$$
where RO$_2$, HO$_2$, NO, NO$_2$ and OH are the concentrations of corresponding species and $k_A$ (A = RO$_2$, HO$_2$,
NO, NO$_2$ and OH) is the reaction rate constant of RO$_2$ with A. For non-acyl RO$_2$, the term $k_{\mathrm{NO_2}}\mathrm{RO_2} \cdot \mathrm{NO_2}$
is not included; for cases with well-known pathways only (RO$_2$+HO$_2$, RO$_2$+RO$_2$, RO$_2$+NO and RO$_2$+NO$_2$;
see Section 3.1), the term $k_{\mathrm{OH}}\mathrm{RO_2} \cdot \mathrm{OH}$ is excluded. $k_{\mathrm{RO_2}}$ needs to be given a value (which may be the
main levels of RO$_2$ self-/cross-reaction rate constants in this study, 1x10$^{-13}$ and 1x10$^{-11}$ cm$^3$ moelcule$^{-1}$ s$^{-}$
$^1$, or other values depending on the RO$_2$ type).
At the steady state, $P$ and $L$ are equal. For an ambient/chamber setting, OH, HO$_2$, NO, NO$_2$ and
OHR$_{VOC}$ are often measured or known. In this case, simultaneously considering Eqs. A1 and A2 yields a
quadratic equation of RO$_2$ concentration (the only unknown). Then generic RO$_2$ concentration can be
easily obtained by solving this equation:
$$\mathrm{RO_2} = \left(-K + \sqrt{K^2 + 8k_{\mathrm{RO_2}} \cdot \mathrm{OHR_{VOC}} \cdot \mathrm{OH}}\right) \big/ \left(4k_{\mathrm{RO_2}}\right) \tag{A3}$$
where $K = k_{\mathrm{HO_2}}\mathrm{HO_2} + k_{\mathrm{NO}}\mathrm{NO} + k_{\mathrm{NO_2}}\mathrm{NO_2} + k_{\mathrm{OH}}\mathrm{OH}$.

**Conflicts of interest**
There are no conflicts to declare.

**Acknowledgements**
This work was partially supported by grants EPA STAR 83587701-0, NSF AGS-1740610, NSF AGS-1822664,
NASA NNX15AT96G, and DOE(BER/ASR) DE-SC0016559. We thank the following individuals for providing



data from atmospheric field studies: Tran Nguyen and Jordan Krechmer (FIXCIT), William Brune (SOAS
and ATom), Pedro Campuzano-Jost (ATom), Daun Jeong and Saewung Kim (GoAmazon). We are also
grateful to John Crounse, Joel Thornton, Paul Ziemann, Dwayne Heard, Paul Wennberg, Andrew Lambe
and William Brune for useful discussions and Donna Sueper for her assistance in the development of
the $RO_2$ Fate Estimator. NCAR is sponsored by the National Science Foundation. EPA has not reviewed
this manuscript and thus no endorsement should be inferred.





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






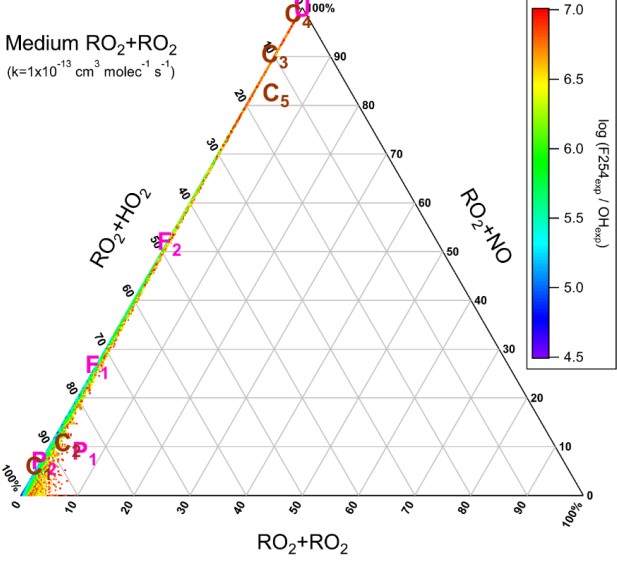


**(a) OFR185**

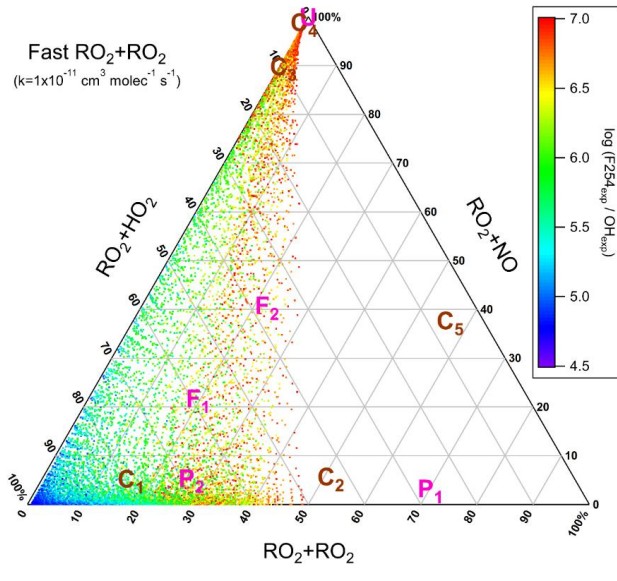


**(b) OFR185**





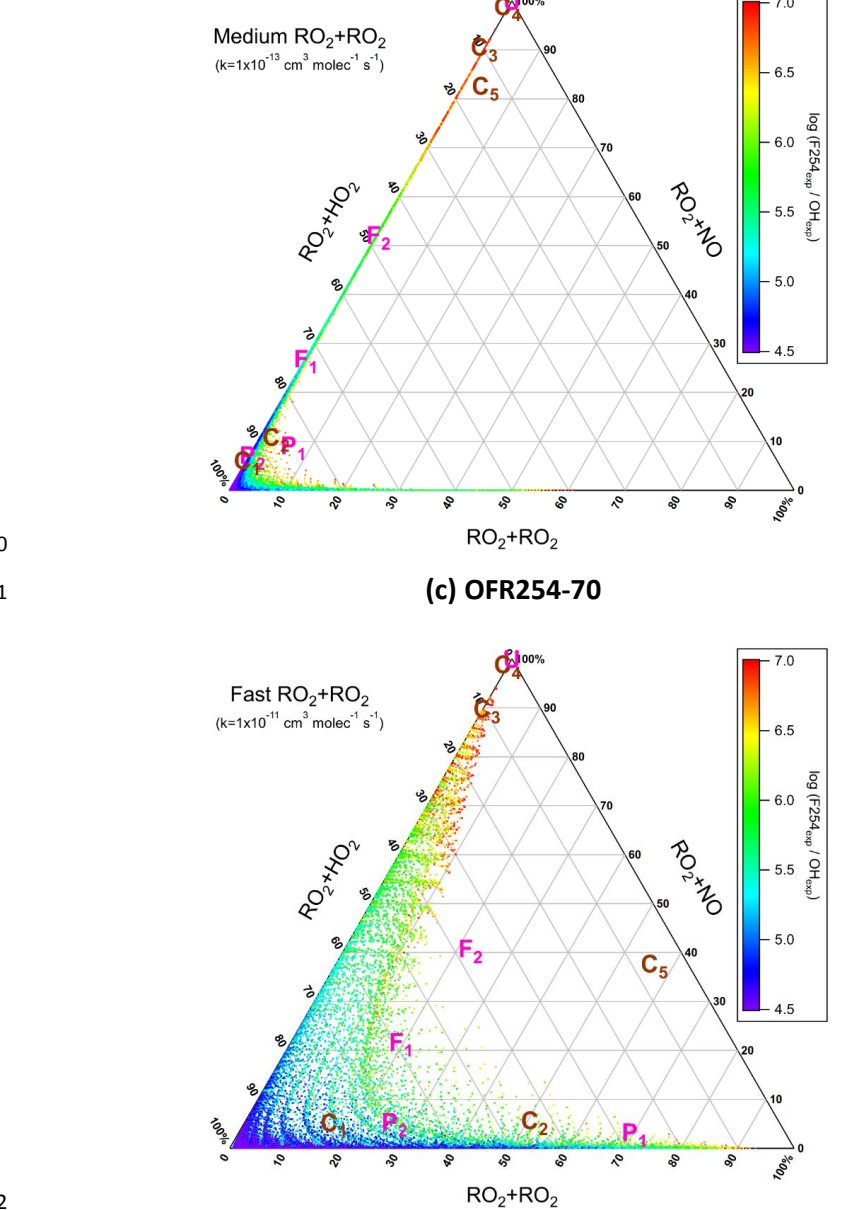

**(c) OFR254-70**

**(d) OFR254-70**

**Figure 1.** Triangle plots of $RO_2$ fate by $RO_2+HO_2$, $RO_2+RO_2$ and $RO_2+NO$ (without $RO_2+OH$ and $RO_2$ isomerization considered in the model) for $RO_2$ with the medium self/cross reaction rate constant ($1\times10^{-13}$ $cm^{-3}$ $molecule^{-1}$ $s^{-1}$) in (a) OFR185 (including OFR185-i$N_2$O) and (c) OFR254-70 (including OFR254-70-i$N_2$O) and for RO2 with the fast self/cross reaction rate constant ($1\times10^{-11}$ $cm^{-3}$ $molecule^{-1}$ $s^{-1}$) in (b) OFR185 (including OFR185-i$N_2$O) and (d) OFR254-70 (including OFR254-70-i$N_2$O). Inclined tick values on an axis indicate the grid lines that should be followed (in parallel to the inclination) to read the





corresponding values on this axis. The OFR data points are colored by the logarithm of the exposure ratio
between 254 nm photon flux and OH, a measure of badness of OFR conditions in terms of 254 nm organic
photolysis. Several typical ambient and chamber cases (see Table 2 for details of these cases) are also
shown for comparison.





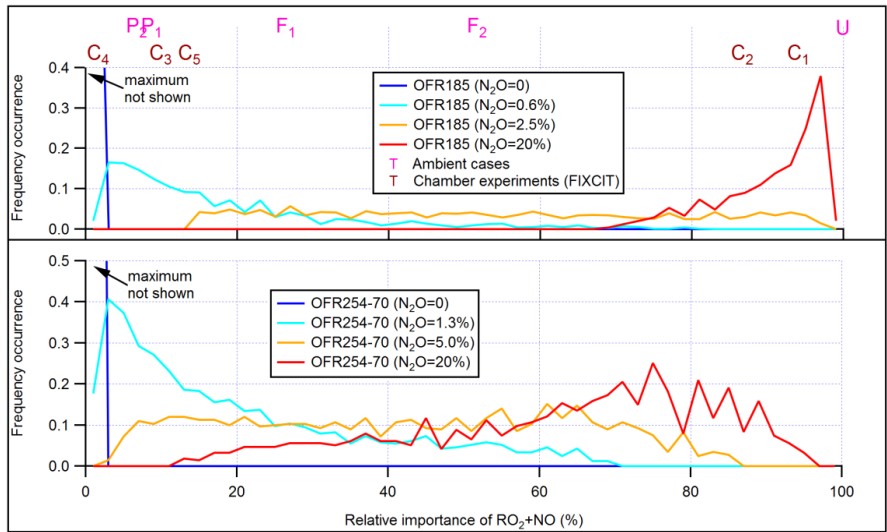

**Figure 2.** Frequency distributions of the relative importance of $RO_2+NO$ in the fate of $RO_2$ (with medium self/cross reaction rate constant and without $RO_2+OH$ and $RO_2$ isomerization considered) for OFR185 (including OFR185-i$N_2$O) and OFR254-70 (including OFR254-70-i$N_2$O). Distributions for several different $N_2$O levels are shown. Only good and risky conditions (in terms of non-tropospheric organic photolysis) are included in the distributions. Also shown is the relative importance of $RO_2+NO$ for several typical ambient and chamber cases (see Table 2 for details of these cases).





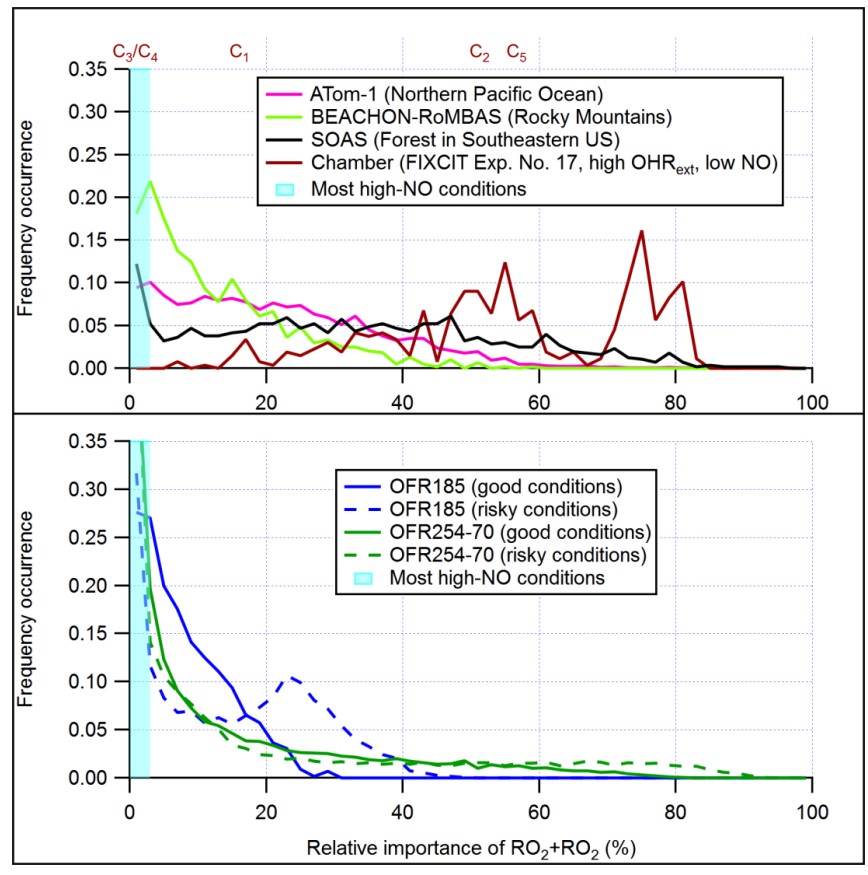

**Figure 3.** Frequency distributions of the relative importance of $RO_2+RO_2$ in the fate of $RO_2$ (with fast self/cross reaction rate constant and without $RO_2+OH$ and $RO_2$ isomerization considered) for OFR185 (including OFR185-i$N_2O$), OFR254-70 (including OFR254-70-i$N_2O$) and a chamber experiment and in the atmosphere (a couple of different environments). The OFR distributions for good and risky conditions (in terms of 254 nm organic photolysis, see Table S1 for the definitions of these conditions) are shown separately. Also shown is the relative importance of $RO_2+RO_2$ for several typical chamber cases (see Table 2 for details of these cases). The range of the $RO_2+RO_2$ relative importance for most high-NO conditions is highlighted in cyan.





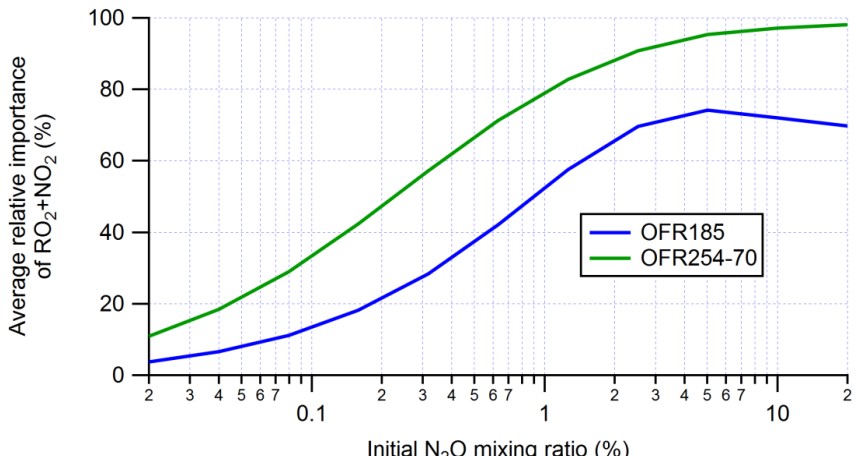

**Figure 4.** Average relative importance of $RO_2+NO_2$ in acyl $RO_2$ fate ($RO_2+OH$ and $RO_2$ isomerization not
considered) in OFR185 (including OFR185-iN$_2$O) and OFR254-70 (including OFR254-70-iN$_2$O). The
averages are calculated based on good and risky conditions (in terms of non-tropospheric organic
photolysis) only.




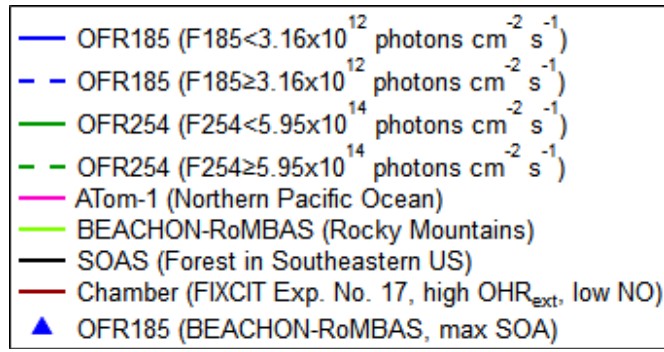

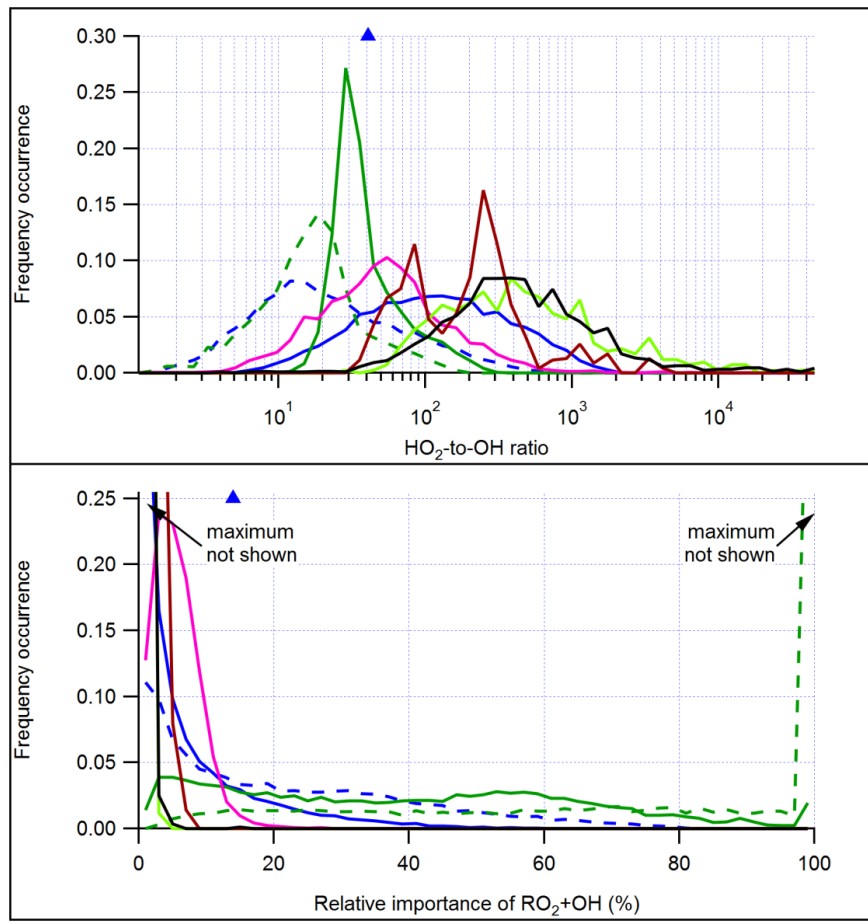

**Figure 5.** Frequency distributions of (top) the HO$_2$-to-OH ratio and (bottom) the relative importance of
RO$_2$+OH in the fate of RO$_2$ (with medium self/cross reaction rate constant) for OFR185 (including OFR185-
iN$_2$O), OFR254-70 (including OFR254-70-iN$_2$O) and a chamber experiment and in the atmosphere (a
couple of different environments). The OFR distributions for lower (F185<3.16x10$^{12}$ photons cm$^{-2}$ s$^{-1}$;
F254<5.95x10$^{14}$ photons cm$^{-2}$ s$^{-1}$) and higher UV (F185≥3.16x10$^{12}$ photons cm$^{-2}$ s$^{-1}$; F254≥5.95x10$^{14}$
photons cm$^{-2}$ s$^{-1}$) are shown separately. Only good and risky conditions (in terms of non-tropospheric
organic photolysis) are included in the distributions for OFRs. Also shown are the HO$_2$-to-OH and the
relative importance of RO$_2$+OH for OFR experiments with ambient air input in field studies.





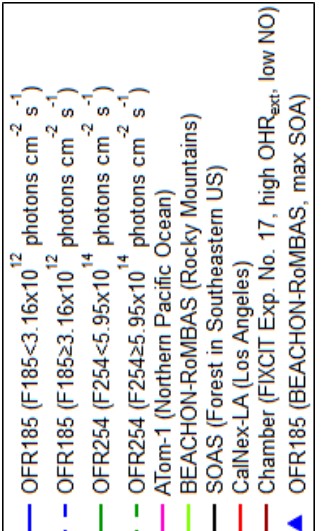

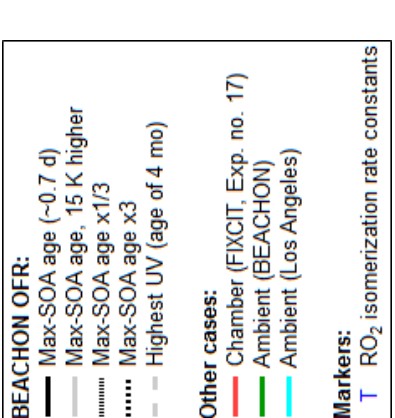

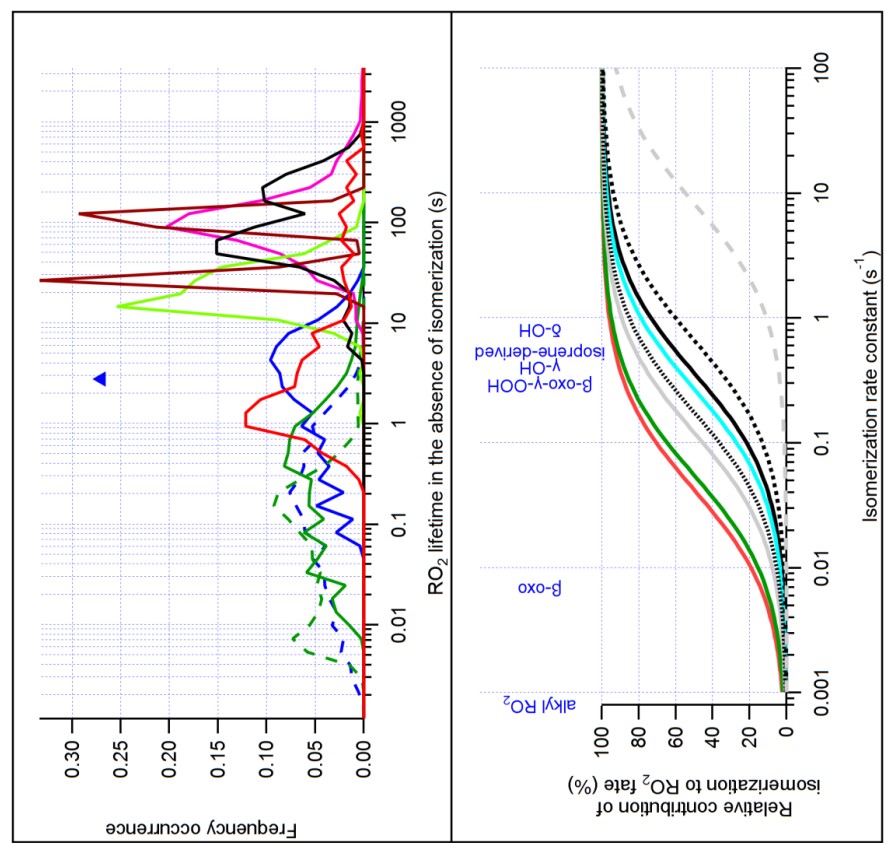



**Figure 6.** (top) Same format as Fig. 5, but for $RO_2$ lifetime ($RO_2$ isomerization included in the model but excluded from lifetime calculation). (bottom) Relative contribution of isomerization to $RO_2$ fate as a function of $RO_2$ isomerization rate constant in several model cases for OFR experiments in the BEACHON-RoMBAS campaign (Palm et al., 2016), in a chamber experiment and in two ambient cases. Isomerization rate constants of several $RO_2$ (Crounse et al., 2013; Praske et al., 2018) are also shown.






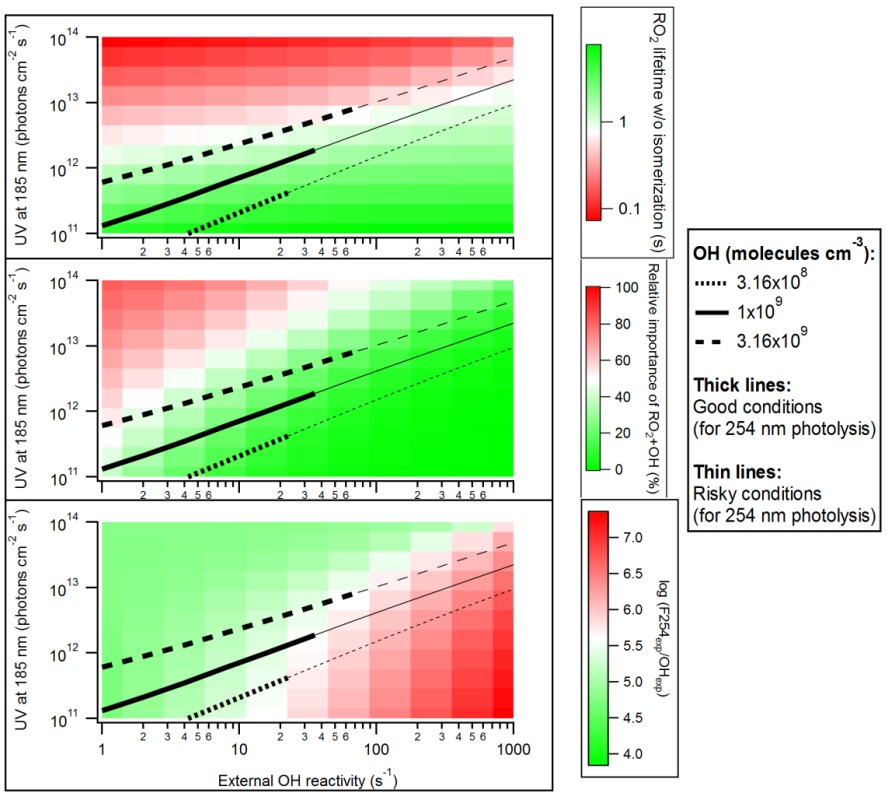

**Figure 7.** (top) $RO_2$ lifetime in the absence of isomerization, (middle) relative importance of $RO_2$+OH in $RO_2$ fate and (bottom) logarithm of the exposure ratio between 254 nm photon flux and OH as a function of 185 nm photon flux and external OH reactivity for OFR185 at $N_2O$=0 and $H_2O$=2.3%. Three lines denoting conditions leading to OH of $3.16 \times 10^8$, $1 \times 10^9$ and $3.16 \times 10^9$ molecules $cm^{-3}$, respectively, are added in each panel. The thick and thin parts of these lines correspond to good and risky conditions (in terms of 254 nm organic photolysis (which is usually worse than 185 nm organic photolysis; Peng et al., 2016) respectively.



**Table 1.** Rate constants [in $cm^3$ molecule$^{-1}$ s$^{-1}$ except for isomerization (in s$^{-1}$)] / cross section (in $cm^2$) and
product(s) of $RO_2$ loss pathways. Only organic species are listed for product(s).

| $RO_2$ loss pathway | Rate constant / cross section | Product(s) |
|---|---|---|
| $RO_2$+$HO_2$ | $1.5 \times 10^{-11}$ [a] | mainly ROOH for most $RO_2$ [a] |
| $RO_2$+NO | $9 \times 10^{-12}$ [a] | RO, $RONO_2$ [b] |
| $RO_2$+$RO_2$ | Primary: ~$10^{-13}$ [a] <br> Secondary: ~$10^{-15}$ [a] <br> Tertiary: ~$10^{-17}$ [a] <br> Substituted: can be up to 2 orders of magnitude higher[b] <br> Acyl: ~$10^{-11}$ [b] | ROH+R(=O), RO+RO, ROOR [a] |
| $RO_2$+$NO_2$ (in OFRs) | $7 \times 10^{-12}$ [c] | $RO_2NO_2$ [b] |
| $RO_2$+OH | $1 \times 10^{-10}$ [d] | ROOOH (for ≥C4 $RO_2$), RO (smaller $RO_2$)[e] |
| $RO_2$ isomerization | Autoxidation: ~$10^{-3}$–$10^2$ [f] <br> Other: up to $10^6$ [g] | generally another $RO_2$ |
| $RO_2$ photolysis | ~$10^{-18}$ at 254 nm[h] <br> ~$10^{-21}$–$10^{-19}$ in UVA and UVB[h] | mainly R, other photochemical products possible[i] |
| $RO_2$+$NO_3$ | ~$1$–$3 \times 10^{-12}$ [b] | RO [b] |
| $RO_2$+$O_3$ | ~$10^{-17}$ [b] | RO [b] |

[a]: Ziemann and Atkinson (2012);
[b]: Orlando and Tyndall (2012);
[c]: typical value within the reported range in Orlando and Tyndall (2012); thermal decomposition rate
constants of nitrates of acyl and non-acyl $RO_2$ are assumed to be 0.0004 and 3 s$^{-1}$, respectively, also
typical values within the reported ranges in Orlando and Tyndall (2012);
[d]: value used in the present work based on Bossolasco et al. (2014); Assaf et al. (2016, 2017a); Müller
et al. (2016); Yan et al. (2016);
[e]: Müller et al. (2016); Yan et al. (2016); Assaf et al. (2017b, 2018);
[f]: Crounse et al. (2013);
[g]: Knap and Jørgensen (2017);
[h]: Burkholder et al. (2015);
[i]: Klems et al. (2015).



**Table 2.** Several typical ambient and chamber (the FIXCIT campaign) cases that are compared to OFR cases.

| Type | Label | Case | $OHR_{VOC}$ (s$^{-1}$) | OH | NO | HO$_2$ |
|---|---|---|---|---|---|---|
| Ambient | P$_1$ | Pristine (Pacific Ocean, high RO$_2$)[a] | 1.9 | 0.39 ppt | 1.9 ppt | 11 ppt |
| | P$_2$ | Pristine (Pacific Ocean, typical)[a] | 1 | 0.25 ppt | 3 ppt | 25 ppt |
| | F$_1$ | Forested (Rocky Mountains)[b] | N/A[c] | 1 ppt | 60 ppt | 100 ppt |
| | F$_2$ | Forested (Amazon, wet season)[d] | 9.6 | $1.2 \times 10^6$ molecules cm$^{-3}$ | 37 ppt | $5.1 \times 10^8$ molecules cm$^{-3}$ |
| | U | Urban (Los Angeles)[e] | 25[f] | $1.5 \times 10^6$ molecules cm$^{-3}$ [g] | 1.5 ppb[i] | $1.5 \times 10^8$ molecules cm$^{-3}$ [g] |
| Chamber (FIXCIT) | C$_1$ | Exp. No. 25[h] | 30.5[i] | $3 \times 10^6$ molecules cm$^{-3}$ | 15 ppt | 150 ppt |
| | C$_2$ | Exp. No. 17[h] | 116[i] | $1.2 \times 10^6$ molecules cm$^{-3}$ | 10 ppt | 50 ppt |
| | C$_3$ | Exp. No. 26[h] | 32[i] | $2 \times 10^7$ molecules cm$^{-3}$ | 3.5 ppb | 230 ppt |
| | C$_4$ | Exp. No. 22[h] | 147[i] | $2.3 \times 10^6$ molecules cm$^{-3}$ | 430 ppb | 4.3 ppb |
| | C$_5$ | Exp. No. 16[h] | 45.7[i] | $4 \times 10^6$ molecules cm$^{-3}$ | 80 ppt | 8 ppt |

[a]: Wofsy et al. (2018) for the Atom-1 Campaign;
[b]: Fry et al. (2013), for the BEACHON-RoMBAS campaign;
[c]: RO$_2$ concentration was given in Fry et al. (2013) (50 ppt), so that OHR$_{VOC}$ is not needed for RO$_2$ fate estimation;
[d]: personal communication from Daun Jeong and Saewung Kim for the GoAmazon Campaign (Martin et al., 2016, 2017);
[e]: typical case in the CalNex-LA campaign (Ryerson et al., 2013);
[f]: estimated (Peng et al., 2016);
[g]: typical ambient value (Mao et al., 2009; Stone et al., 2012);
[h]: data from Nguyen et al. (2014);
[i]: initial value.