# Peer review of "Organic peroxy radical chemistry in oxidation flow reactors and environmental chambers"

_Atmospheric Chemistry and Physics, 2018_

## Referee Comment (RC1) · Anonymous Referee #1 · 22 Oct 2018

This paper follows a number of others reporting the characterisation and optimisation of operating conditions for oxidation flow reactors (OFRs) through both experimental and modeling studies. This work focuses on a model study of the fate of organic peroxy radicals within OFRs under different operating conditions and makes comparisons to the fates of such species in the atmosphere. While the rationale for such a study is sound, and the methods described are appropriate, the paper is quite long for the information it contains, and there are questions as to the wider interest and novelty of the work. It would help if the authors could state the main scientific outcomes and objectives of this work more clearly, and if some detail could be provided which outlines how assumptions regarding the fates of RO2 species have potentially impacted the

results of previous studies.

Specific comments are listed below.

Page 4, lines 117-122: Is production of RO2 from ozonolysis reactions or reactions between O(1D) and VOCs considered for conditions when significant ozone/O(1D) are present? Although photolysis of organics is considered, is there any consideration of photolysis of oxygenated VOCs (formaldehyde or acetaldehyde for example) which may photolyse to generate RO2 radicals?

Page 4, lines 133-134: The authors assume average ambient HO2 concentrations of $1.5 \times 10^8$ molecules cm-3 and kHO2+RO2 of $1.5 \times 10^{-11}$ cm3 molecule-1 s-1. The generic rate coefficient for HO2 + RO2 used seems high, for HO2 + CH3O2 the rate coefficient is $5.2 \times 10^{-12}$ cm3 molecule-1 s-1, while that for HO2 + C2H5O2 is $6.9 \times 10^{-12}$ cm3 molecule-1 s-1. How do the assumptions regarding [HO2] and kHO2+RO2 influence the results reported in this work? Similarly, how does the assumption regarding kRO2+NO influence the results?

Page 4, line 152 (and elsewhere): Please consider changing 'RO2s' to 'RO2 radicals' or similar.

Page 5, line 175: Please quantify the statement 'acylperoxy nitrates barely decompose' with an example.

Page 6, line 205: What is the rationale for this production rate of OH? What is the VOC concentration used? (i.e. What is the pseudo-first-order rate coefficient?)

Page 6, line 220 and page 10, line 370: Are the results from these simulations reported anywhere? How is this estimate achieved?

Page 8, line 304: I'm not sure forested areas should be described as 'low VOC' given high biogenic emissions in such regions.

Page 9, line 307: Are the labels C1, C2 etc. shown in the Figures described anywhere?

[Figure]

Page 12, line 421: Subscript in 'RO2'.

Page 13, line 490: Should this read ' Neither is the fast RO2 + RO2 . . .'?

Page 16, line 585: Please provide a reference to the statement '. . . other major gas-phase radical reactions have weak or no temperature-dependence' or compare to a typical change in rate coefficient over a similar temperature range for RO2 + NO, RO2 + HO2 and RO2 + RO2.

Page 16, lines 599-611: The utility of the RO2 fate estimator is unclear. What does it do above and beyond a simple yield/budget calculations requiring knowledge of [HO2], [RO2], [NO] and the corresponding rate coefficients? It would be surprising if groups performing OFR studies, or similar, weren't already able to do such calculations.

Figure 1: The labels C1, P1, U etc. are unclear and/or overlapping with other labels on the plots.

---

## Referee Comment (RC2) · Anonymous Referee #2 · 29 Oct 2018

This manuscript describes the use of a chemical model to evaluate the use of oxidative flow reactors. Overall, the work is important for groups using oxidative flow reactors, and includes a high quality analysis of the chemistry in those flow reactors, and should be published subject to appropriate revision. However, I have a few qualms that should be addressed by the authors, noted below.

It is only a handful of groups that use OFRs, and the authors should address the generalizability of their research. Aside from acting as a handbook for OFR users, are there general notes on tropospheric chemistry that the authors can provide to the reader? For example, the relative importance of the different fates of RO2 is generally

[Figure]

interesting, and a clear summarizing point from that first figure could be of use and generally interest.

The authors discuss RO2 as a general radical term, and in general their analysis makes sense from that perspective. However, in high NOx environments, RO2+NO dominantly produces RO+NO2, but sometimes produces organic nitrates (RONO2). The frequency of this branching will depend on the chemical identity of the RO2 precursor. I suspect this chemistry will impact the rates and radical balance in the OFR+N2O experiments. The authors include this reaction in Table 1, but do not discuss this reaction at all. Their model should be able to use RONO2 species to track the RO2 fate in high NOx experiments and see if the values are tropospherically relevant (i.e. will the OFR model - or OFR itself - produce a branching ratio that matches previous laboratory or field experiments?).

Finally, I would like to see the 'Guidelines for OFR Operation' either have a short bullet point summary of key points (or those in the Conclusions section), or be made more concise. Overall, it would behoove the authors to consider whether all the text and figures/tables are necessary to make their main points, or if there are additional places that could be removed. The paper is dense, which will reduce the readership. Reducing the number of acronyms (and making a table of whatever acronyms are left) would be very helpful for readability. There are so many 'OFR-subversions' that I had a challenging time reviewing portions of the manuscript.

Technical comments

line 51: remove the extra "("

line 212: I genuinely don't understand this sentence - please clarify (i.e. an accuracy of what?)

line 352: Acyl RO2 +NO2 is typically referred to as an 'equilibrium', not 'quasi-irriversible reaction'. Consider what happens as temperature is increased - in the

**[ACPD]{.blue}**

Interactive
comment

troposphere in summer, this equilibrium is important for most PAN-type compounds, and cannot be ignored! If this is the case in most OFRs, then there is a more serious problem with the RO2/NO2 and NO/NO2 ratios...

line 361: what are typical NO/NO2 ratios in the OFR and in the troposphere? It would be helpful to summarize in a sentence.

line 371: The sentence that states that acyl RO2 dominate aged air plumes requires a reference.

line 432: 's' on the end of 'reaction(s)' should be deleted

———————————————

---

## Author Comment (AC1) · 14 Dec 2018

**Response to Reviews for the ACPD paper "Organic peroxy radical chemistry in oxidation flow reactors and environmental chambers and their atmospheric relevance"**

We thank the referees for their reviews. To facilitate the review process, we have copied the reviewer comments in black text. Our responses are in regular blue font. We have responded to all the referee comments and made alterations to our paper (**in bold text**). Figures, tables, and sections in the responses are numbered as in the *revised* manuscript unless otherwise specified, while page and line numbers refer to the ACPD paper.

**Anonymous Referee #1**

This paper follows a number of others reporting the characterisation and optimisation of operating conditions for oxidation flow reactors (OFRs) through both experimental and modeling studies. This work focuses on a model study of the fate of organic peroxy radicals within OFRs under different operating conditions and makes comparisons to the fates of such species in the atmosphere.

R1.0) While the rationale for such a study is sound, and the methods described are appropriate, the paper is quite long for the information it contains, and there are questions as to the wider interest and novelty of the work. It would help if the authors could state the main scientific outcomes and objectives of this work more clearly, and if some detail could be provided which outlines how assumptions regarding the fates of RO2 species have potentially impacted the results of previous studies.

We believe that our paper fulfills an important need for the rapidly growing and interdisciplinary OFR research community. The paper provides a critical assessment of best practices in the use of OFRs, and dispels common notions regarding the shortcomings of OFRs, and thus is useful. We have added the following text to the introduction (after L67 of the ACPD version) to clarify this matter:

**"The use of oxidation flow reactors is growing rapidly in the atmospheric chemistry community. Some researchers have raised two concerns with regard to OFRs: (1) the chemical regime of OFRs may be unrealistic compared to the atmosphere and (2) OFRs are derivative of flow reactors with a long tradition in atmospheric chemistry, especially for chemical kinetic measurements, and thus there is not much new to be discussed or analyzed in their chemistry. While it is true that OFRs follow the tradition of flow tubes used in atmospheric chemistry, they attempt to simulate a much more complex system all-at-once and typically use much longer residence times, and thus many fundamental and practical issues arise that have not been addressed before. The need to achieve longer effective photochemical ages within a short residence time can, however, lead to**

**the occurrence of undesirable oxidation pathways. This paper uses computer modeling to define useful ranges in which to work."**

Therefore, we strongly believe that the present paper does address the needs of people interested in practical OFR application and those interested in the study of fundamental chemical pathways). We do make a reasonable attempt to present the methods and results in a way that is both rigorous and accessible to many OFR users with limited chemical knowledge (e.g. researchers with more of an aerosol and/or engineering training), even though to knowledgeable chemists this paper might appear to be somewhat wordy and detail-oriented.

In the response to R1.8, we have modified some text in the ACPD paper to give more details on and highlight several features of $RO_2$ chemistry in the atmosphere and chambers.

See also the response to comment R1.12.

R1.1) Page 4, lines 117-122: Is production of RO2 from ozonolysis reactions or reactions between O(1D) and VOCs considered for conditions when significant ozone/O(1D) are present? Although photolysis of organics is considered, is there any consideration of photolysis of oxygenated VOCs (formaldehyde or acetaldehyde for example) which may photolyse to generate RO2 radicals?

While $RO_2$ production from ozonolysis and photolysis of VOCs and their reactions with $O(^1D)$ is possible in the OH OFRs that are the focus of this paper, these non-OH pathways are significant only when OH is suppressed or not sufficiently produced, i.e. under "risky" or "bad" conditions as defined in the paper. We have previously shown that all these non-OH reactions become important for similar reasons and photolysis at 254 nm is usually the most significant type of non-OH reactions (Peng et al., 2016). The physical conditions leading to significant 254 nm VOC photolysis (non-tropospheric chemistry) are of little experimental interest. Thus we do not believe that it is necessary to include $RO_2$ production through non-OH pathways in this study.

We have added the following text to the end of the paragraph between L117 and L125:

"**$RO_2$ production through other pathways, e.g. VOC ozonolysis and photolysis, is not considered, since all non-OH pathways of VOC destruction only become significant at low $H_2O$ and/or high $OHR_{ext}$ (Peng et al., 2016). These conditions lead to significant non-tropospheric VOC photolysis and thus are of little experimental interest.**"

R1.2) Page 4, lines 133-134: The authors assume average ambient HO2 concentrations of $1.5x10^8$ molecules cm-3 and kHO2+RO2 of 1.5x10-11 cm3 molecule-1 s-1. The generic rate coefficient for HO2 + RO2 used seems high, for HO2 + CH3O2 the rate coefficient is 5.2x10-12 cm3 molecule-1 s-1, while that for HO2 + C2H5O2 is 6.9x10-12 cm3 molecule-1 s-1. How do

the assumptions regarding [HO2] and kHO2+RO2 influence the results reported in this work? Similarly, how does the assumption regarding kRO2+NO influence the results?

In the typical OFR experiments focused on SOA formation that we are mainly studying in this paper, $CH_3O_2$ and $C_2H_5O_2$ are minor contributors to the total $RO_2$ pool. Their formation rates through methane and ethane oxidation are very small compared to the formation of larger $RO_2$ radicals from other VOCs. We thus do not believe that $CH_3O_2$ and $C_2H_5O_2$ are important intermediates of VOC oxidation that are able to significantly alter the overall OH, $HO_2$, and $RO_2$ budget in OFRs. In the ambient and chamber cases, OH and $HO_2$ have been prescribed. If $CH_3O_2$ and $C_2H_5O_2$ are not the $RO_2$ of interest (which is usually the case in SOA formation studies), the different rate constants of the reactions of $CH_3O_2$ and $C_2H_5O_2$ with $HO_2$ than the typical value used for $RO_2+HO_2$ in this study will have no impact on the results of the ambient and chamber cases.

For other unsubstituted and oxygenated $RO_2$ radicals, the rate constants of their reactions with $HO_2$ are indeed around $1.5 \times 10^{-11}$ cm$^3$ molecule$^{-1}$ s$^{-1}$ ($\sim 1-2 \times 10^{-11}$ cm$^3$ molecule$^{-1}$ s$^{-1}$; see Table 5 of Orlando and Tyndall, 2012). And the rate constants of $RO_2+NO$ are indeed very close to $9 \times 10^{-12}$ cm$^3$ molecule$^{-1}$ s$^{-1}$ (see Table 1 of Orlando and Tyndall, 2012) for most $RO_2$ radicals, including $CH_3O_2$ and $C_2H_5O_2$. Only the rate constants of acyl $RO_2+NO$ are $\sim$ x2 the value used in the paper.

For the modified text to clarify this issue, please refer to the response to comment R2.2.

R1.3) Page 4, line 152 (and elsewhere): Please consider changing 'RO2s' to 'RO2 radicals' or similar.

We have changed "$RO_2$s" to "$RO_2$ radicals" throughout the paper.

R1.4) Page 5, line 175: Please quantify the statement 'acylperoxy nitrates barely decompose' with an example.

We have modified the relevant sentences in L175–176 with some detail added to clarify this:

"**In OFRs operated at room temperature, acylperoxy nitrates barely decompose, as their thermal decomposition lifetime is typically ~1 h (Orlando and Tyndall, 2012), while OFR residence time is usually a few minutes. In contrast, peroxy nitrates of non-acyl RO$_2$ do decompose on a timescale of 0.1 s (Orlando and Tyndall, 2012).**"

R1.5) Page 6, line 205: What is the rationale for this production rate of OH? What is the VOC concentration used? (i.e. What is the pseudo-first-order rate coefficient?)

First, we would like to clarify that the text in L205 and below discusses OH loss rather than OH production. In this study, we use a proxy of external OH reactant ($SO_2$), but external OH reactivity ($OHR_{ext}$, the pseudo-first-order rate constant of OH loss due to external species (e.g. VOCs, CO, and $SO_2$)) is not constrained to one value. Instead, its effects are explored over a very wide range (1–1000 $s^{-1}$) across our model cases. The initial $SO_2$ concentration used in each model case is determined by the initial $OHR_{ext}$ chosen for that case.

We use $SO_2$ as a surrogate of external OH reactants for simplicity. $OHR_{ext}$ for VOCs evolves over time (upon oxidation of VOCs and the formation and later oxidation of the stable reaction products) (Nehr et al., 2014; Fuchs et al., 2017; Sato et al., 2017; Schwantes et al., 2017). This evolution plays an important role in OH loss over time, but is not well modeled even with explicit chemical scheme such as Master Chemical Mechanism (Sato et al., 2017; Schwantes et al., 2017). Using a slow-reacting external OH reactant, i.e. $SO_2$, as a proxy can roughly account for the relatively slow decay of $OHR_{ext}$ (compared to that of primary VOCs) due to the generation of second and later generation products. As we already stated in the ACPD paper (L208–209), this rationale has been discussed in detail in our previous papers (Peng and Jimenez, 2017; Peng et al., 2018). As this paper is already long, we prefer not to extensively discuss this approximation in the text again.

Nevertheless, for more clarity, we have modified the text in L205 to read:

"**A generic slow-reacting VOC (with the same OH rate constant as $SO_2$) is used as the external OH reactant. Its initial concentration is determined by the initial $OHR_{ext}$ in each model case. Then as this proxy external OH reactant slowly reacts, $OHR_{ext}$ slowly decays. This slow change in $OHR_{ext}$ represents not only the decay of the initial reactant but also the generation and consumption of later-generation products that continue to react with OH. The reason for this approximation has been discussed in detail in previous OFR modeling papers (Peng and Jimenez, 2017; Peng et al., 2018).**"

R1.6) Page 6, line 220 and page 10, line 370: Are the results from these simulations reported anywhere? How is this estimate achieved?

To address the Referee's question, we have modified text to L219 and L369 to include more details about these calculations. The modified text in L219 reads as follows:

"**We used the fully chemically explicit (automated chemical mechanism generation based on available knowledge) box-model GECKO-A (Aumont et al., 2005) to simulate OH oxidation of several simple VOCs (e.g. propane and decane) under various OFR conditions with zero-NO. We consistently find that β~0.3.**"

And the modified text in L369:

"**However, simulations using the GECKO-A model in urban (Mexico City) and forested (Rocky Mountains) atmospheres (Figure S8) show that acyl RO$_2$ can still be a major (very roughly 1/3) component of RO$_2$ at ages of several hours or higher.**

[Figure]

[Figure]

**Figure S8. (a) RO$_2$ concentration and composition [primary (1ry), secondary (2ry), tertiary (3ry) and acyl RO$_2$] as a function of aging time for the simulation of a parcel of air advected from Mexico City during the MILAGRO 2006 campaign using the fully explicit GECKO-A model (Lee-Taylor et al., 2015). (b) The same for a GECKO-A simulation of air in a Rocky Mountain pine forest for the average diurnal cycle during the BEACHON-RoMBAS 2011 campaign (Palm et al., 2016; Hunter et al., 2017). Nighttime is denoted by shaded area.**"

R1.7) Page 8, line 304: I'm not sure forested areas should be described as 'low VOC' given high biogenic emissions in such regions.

For more clarity, we have modified the text to L303 to read:

"**Although OFRs can reasonably reproduce RO$_2$ fates in typical low- and moderate-OHR$_{ext}$ ambient environments (e.g. typical pristine and forested areas; Figs. 1b,d and 3) and low-OHR$_{ext}$ chambers, OFR185 cannot achieve relative importance of RO$_2$+RO$_2$ significantly larger than 50%, such as found in remote environments with higher VOC (e.g. P1 in Fig.**

**1) and high-OHR$_{ext}$ chamber experiments (e.g. C2 and C5 in Fig. 1; the distribution for C2 is also shown in Fig. 3).**"

R1.8) Page 9, line 307: Are the labels C1, C2 etc. shown in the Figures described anywhere?

We have described the typical cases corresponding to these labels in Section 3.1.1 of the ACPD paper. Nevertheless, for added clarity we have modified the paragraph starting from L262 to include more discussion about the typical ambient and chamber cases:

"**In this case non-acyl RO$_2$ can have only three fates, i.e. RO$_2$+HO$_2$, RO$_2$+NO and RO$_2$+RO$_2$. The relative importance of these three fates can be shown in a triangle plot (Figure 1). The figure includes data points of OFR185 (including OFR185-iN$_2$O) and OFR254-70 (including OFR254-70-iN$_2$O), as well as several typical ambient and chamber studies, including two pristine remote area cases (P$_1$ and P$_2$) from the ATom-1 study (Wofsy et al., 2018), two forested area cases (F$_1$ and F$_2$) from the BEACHON-RoMBAS and GoAmazon campaigns, respectively (Ortega et al., 2014; Martin et al., 2016, 2017), an urban area case (U) from the CalNex-LA campaign (Ryerson et al., 2013) and five typical chamber experiment cases (C$_1$–C$_5$) from the FIXCIT study (Nguyen et al., 2014). These typical cases shown in Fig. 1 bring to light several interesting points:**

- **In all ambient and chamber cases, medium and slower RO$_2$+RO$_2$ contribute negligibly to the RO$_2$ fate. This confirms a common impression that self-/cross-reactions of many RO$_2$ radicals do not significantly affect RO$_2$ fates.**
- **However, if RO$_2$ self-/cross-reacts rapidly, RO$_2$+RO$_2$ can be the most important loss pathway among RO$_2$+RO$_2$, RO$_2$+HO$_2$ and RO$_2$+NO even in pristine regions with higher VOC (e.g. P$_1$ in Fig. 1) compared to an average pristine region case (P$_2$). Note that the P$_1$ case is still very clean compared to typical forested and urban areas (Table 2).**
- **Forested areas located in the same region as pollution sources are not as "low-NO" as one may expect (points F$_1$ and F$_2$ in Fig. 1). RO$_2$+NO contributes ~20–50% to RO$_2$ loss, as NO and HO$_2$ concentrations are on the same order of magnitude in these cases.**
- **RO$_2$+NO dominates over RO$_2$+RO$_2$ and RO$_2$+HO$_2$ in almost all urban areas. Even in relatively clean urban areas such as Los Angeles during CalNex-LA in 2010 (point U in Fig. 1), average NO is ~1 ppb, still sufficiently high to ensure the dominance of RO$_2$+NO among the three pathways.**
- **Various chamber cases in the FIXCIT campaign (low to high OHR$_{ext}$; low to high NO; points C$_x$ in Fig. 1) are able to represent specific RO$_2$ fates that appear in different regions in the atmosphere.**

**On these triangle plots, points for bad OFR conditions (in terms of non-tropospheric photolysis) are not shown because of the lack of experimental interest…**"

R1.9) Page 12, line 421: Subscript in 'RO2'.

We have corrected it as suggested by the Referee.

R1.10) Page 13, line 490: Should this read 'Neither is the fast RO2 + RO2 …'?

We quote relevant sentences in that paragraph below:

"Since $RO_2+HO_2$ and $RO_2+NO$ both can vary from negligible to dominant $RO_2$ fate in OFRs, chambers and the atmosphere (Figs. 1 and 2), these two pathways are not a concern in OFR atmospheric relevance considerations. Neither is the $RO_2+RO_2$ a major concern. Medium or slower $RO_2+RO_2$ is minor or negligible in the atmosphere and chambers, as well as in OFRs, as long as high $OHR_{ext}$ is avoided in OFR254 (Fig. S2). Fast $RO_2+RO_2$ is somewhat less important in OFRs than in the atmosphere (Figs. 1b,d and 3), but is still qualitatively atmospherically relevant, given the uncertainties associated with the $HO_x$ recycling ratios of various reactive systems and the huge variety of $RO_2$ types (and hence $RO_2+RO_2$ rate constants)."

We do not believe that the relevant text needs to be modified as suggested by the Referee, as the sentence "Neither is the $RO_2+RO_2$ a major concern" is followed by discussions on both medium/slower $RO_2+RO_2$ and fast $RO_2+RO_2$. Both types of $RO_2+RO_2$ are generally atmospherically relevant in OFRs.

R1.11) Page 16, line 585: Please provide a reference to the statement '. . . other major gas-phase radical reactions have weak or no temperature-dependence' or compare to a typical change in rate coefficient over a similar temperature range for RO2 + NO, RO2 + HO2 and RO2 + RO2.

We have added several examples and corresponding references into this sentence. The modified text now reads:

"**A 15 K temperature increase in OFRs would lead to $RO_2$ isomerization being accelerated by a factor of ~3, while other major gas-phase radical reactions have weak or no temperature-dependence (e.g. ~7%, ~5%, ~6% and ~19% slow-downs for isoprene+OH, toluene+OH, typical $RO_2$+NO and $RO_2$+$HO_2$, respectively; Atkinson and Arey, 2003; Ziemann and Atkinson, 2012).**"

R1.12) Page 16, lines 599-611: The utility of the RO2 fate estimator is unclear. What does it do above and beyond a simple yield/budget calculations requiring knowledge of [HO2], [RO2], [NO]

and the corresponding rate coefficients? It would be surprising if groups performing OFR studies, or similar, weren't already able to do such calculations.

Actually, we developed the RO$_2$ Fate Estimator partly due to a request by a well-known atmospheric chemist who is an expert in chamber experiments. While many chamber experimentalists are aware of the importance of RO$_2$ fate in their laboratory experiments, often analyses are presented that are not completely consistent with each other, or that do not include all the relevant pathways. We also found relatively few available datasets in the literature for chamber experiment RO$_2$ fate analysis, and we recommend performing such an analysis or reporting relevant data for it in the paper. Having a standardized RO$_2$ fate estimator available may facilitate some of these tasks.

For OFR users, their awareness of the importance of RO2 fate analysis may generally be lower than chamber experimentalists, as many of these groups have aerosol science or engineering backgrounds (e.g. aerosol optics and emission control of sources such as motor vehicles). As a result, RO$_2$ fate has rarely been reported in OFR studies. A user-friendly tool for these OFR practitioners to analyze this relatively complex problem would be very useful. Also quantities such as HO$_2$ concentration in OFRs are indeed very hard to measure, and many OFR users do not have tools to assess them independently. The estimation equations for these quantities embedded in the OFR RO$_2$ Fate Estimator makes realistic analyses of RO$_2$ fate in OFRs possible.

Therefore, we believe that both of our RO$_2$ Fate Estimators are of great practical interest and do not modify this paragraph (L599–611).

See also the response and added text in response to comment R1.1.

R1.13) Figure 1: The labels C1, P1, U etc. are unclear and/or overlapping with other labels on the plots.

We have modified all triangle plots to avoid overlap of the case labels. Below is an example (Figure                                                                                                                1a).

[Figure]

**Anonymous Referee #2**

This manuscript describes the use of a chemical model to evaluate the use of oxidative flow reactors. Overall, the work is important for groups using oxidative flow reactors, and includes a high quality analysis of the chemistry in those flow reactors, and should be published subject to appropriate revision. However, I have a few qualms that should be addressed by the authors, noted below.

R2.1) It is only a handful of groups that use OFRs, and the authors should address the generalizability of their research. Aside from acting as a handbook for OFR users, are there general notes on tropospheric chemistry that the authors can provide to the reader? For example, the relative importance of the different fates of RO2 is generally interesting, and a clear summarizing point from that first figure could be of use and generally interest.

We would like to clarify that OFRs are not only used by a handful groups. Currently there are ~50 research groups worldwide that use OFRs and this number is increasing very rapidly (probably by 10–15 groups per year). According to Google Scholar, the annual number of publications concerning OFRs has reached ~1/3 of that concerning traditional chambers and is increasing exponentially. We can comfortably claim that OFR has already become a mainstream atmospheric chemistry research tool. A study focused on such a tool, while comparing with the traditional tool (chambers) and the atmosphere, has enough scientific interest and practical importance to stand on its own.

Nevertheless, to address the Referee's concern, we have modified the text in Section 3.1.1 to provide more details of $RO_2$ fates in the troposphere and chambers. Please refer to the response to comment R1.8 for the modified text.

R2.2) The authors discuss RO2 as a general radical term, and in general their analysis makes sense from that perspective. However, in high NOx environments, RO2+NO dominantly produces RO+NO2, but sometimes produces organic nitrates (RONO2). The frequency of this branching will depend on the chemical identity of the RO2 precursor. I suspect this chemistry will impact the rates and radical balance in the OFR+N2O experiments. The authors include this reaction in Table 1, but do not discuss this reaction at all. Their model should be able to use RONO2 species to track the RO2 fate in high NOx experiments and see if the values are tropospherically relevant (i.e. will the OFR model - or OFR itself - produce a branching ratio that matches previous laboratory or field experiments?).

In the model, we focus on simulating generic $RO_2$ concentration and do not explicitly consider subsequent reactions of the products of $RO_2$ loss pathways. The overwhelming majority of $NO_x$ in OFR-i$N_2$O exists in the form of $NO_2$ and $NO_2$ is dominantly produced from the oxidation of NO by $O_3$, $HO_2$ and OH (Peng et al., 2018), whose concentrations are orders of magnitude higher than corresponding ambient values. In the ACPD paper, we have already shown that $RO_2$ concentration in OFRs cannot be enhanced as much as $O_3$, $HO_2$ and OH. Therefore, whether a minor fraction of NO reacted with $RO_2$ produces $NO_2$ or not has virtually no impact on $NO_2$ concentration in OFR-i$N_2$O. Also, the $HO_2$ production from the reaction of RO with $O_2$ is already implicitly accounted for by the $HO_x$ recycling described by β (see Section 2.3). In theory the $RONO_2$ formation branching ratio may affect β, but this impact should be small compared to the total $HO_x$ recycling and generally negligible compared to the total $HO_x$ production (Peng et al., 2015). As a result, subsequent reactions of the products of $RO_2$+NO and their branching ratios do not significantly affect the $HO_x$ and $NO_x$ balances in the simulations of OFR-i$N_2$O.

For simplicity, we do not specify the branching ratios of the RO+NO$_2$ and RONO$_2$ channels. As long as the relative contribution of RO$_2$+NO to the fate of RO$_2$ of interest in OFRs is close to that in other chamber or field experiments, the fractions (yields) of RO and RONO$_2$ in the total amount of the products of RO$_2$ loss pathways will also be close to those in other chamber or field experiments. Practically, readers can easily obtain those yields by multiplying the relative contribution of RO$_2$+NO by the branching ratios for RO$_2$ of interest.

For added clarity, we have modified the text to L145 to read:

"**Recommended general rate constants are available for RO$_2$+HO$_2$ and RO$_2$+NO (Ziemann and Atkinson, 2012; Table 1), albeit with some small dependencies on the type of RO$_2$ and a few deviations that are slightly larger but not important for the overall chemistry (e.g. CH$_3$O$_2$ and C$_2$H$_5$O$_2$ for RO$_2$+HO$_2$). We use these recommended values for generic RO$_2$ in this study. RO$_2$+NO has two main product channels, i.e. RO+NO$_2$ and RONO$_2$, whose branching ratios are RO$_2$-structure-dependent (Ziemann and Atkinson, 2012). We do not include these product channels in this study, since they have negligible impacts on the chemical scheme described here. This feature results from two facts: i) we focus on the generic RO$_2$ and do not explicitly consider the chemistry of products of the different RO$_2$ loss pathways; ii) the channel producing RO and NO$_2$ contributes little to NO$_2$ production (Peng et al., 2018).**"

R2.3) Finally, I would like to see the 'Guidelines for OFR Operation' either have a short bullet point summary of key points (or those in the Conclusions section), or be made more concise. Overall, it would behoove the authors to consider whether all the text and figures/tables are necessary to make their main points, or if there are additional places that could be removed. The paper is dense, which will reduce the readership. Reducing the number of acronyms (and making a table of whatever acronyms are left) would be very helpful for readability. There are so many 'OFR-subversions' that I had a challenging time reviewing portions of the manuscript.

We think that in Section 3.3 there is an obvious logical flow linking the points discussed, and hence prefer to make the bullet point summary in Section 4 (Conclusions) instead of Section 3.3. The modified second paragraph (starting from L630) now reads:

"**Besides the above-mentioned well-known pathways, RO$_2$+OH and RO$_2$ isomerization may also play an important role in RO$_2$ fate and sometimes result in atmospherically irrelevant RO$_2$ chemistry in OFRs. Here we summarize the main findings about all the pathways and the related guidelines for OFR operation:**

- **Under typical high-NO conditions, RO$_2$+NO dominates RO$_2$ fate and RO$_2$ lifetime is too short to allow most RO$_2$ isomerizations, regardless of whether in the atmosphere, chambers or OFRs, thus raising no concern about the atmospheric relevance of the OFR RO$_2$ chemistry.**

- **Under low-NO conditions, OFR254 cannot yield any physical conditions leading to sufficiently long $RO_2$ lifetime for its isomerization because of the high radical levels and their resilience to external perturbations in OFR254.**
- **In OFR185 with strong OH production (and hence high OH), $RO_2$+OH and $RO_2$ isomerization may strongly deviate from that in the atmosphere [becoming important and negligible, respectively, for relatively rapidly isomerizing $RO_2$ (rate constants on the order of 0.1 s$^{-1}$)].**
- **To attain both atmospherically relevant VOC and $RO_2$ chemistries, OFR185 requires high $H_2O$, low UV and low $OHR_{ext}$. These conditions ensure minor or negligible $RO_2$+OH and a relative importance of $RO_2$ isomerization in $RO_2$ fate in OFRs within a factor of ~2 of that in the atmosphere.**
- **Under conditions allowing both VOC and $RO_2$ chemistries to be atmospherically relevant, the maximal photochemical age that can be reached is limited to a few eq. days. This age roughly covers the period required for maximum SOA formation in ambient air.**
- **To most realistically study much higher ages for SOA functionalization/fragmentation by heterogeneous oxidation, a sequence of low-UV SOA formation followed by a high UV condition (in the same reactor or in cascade reactors) may be needed.**
- **High $H_2O$, low UV and low $OHR_{ext}$ in the OFR185-iNO mode can achieve conditions relevant to clean urban atmosphere, i.e. high-NO but not sufficiently high to inhibit common $RO_2$ isomerization.**"

In addition, we have added a glossary table as a part of the appendices to clarify the meanings of the different acronyms. Unfortunately, there are indeed multiple ways of running OFRs, each having very different chemical properties. Users have to choose a specific mode for a given experiment. The mode acronyms are thus necessary when discussing the different modes and their advantages and disadvantages:

"**Appendix A: Glossary of the acronyms (except field campaign names) used in the paper**

| | |
|---|---|
| **OFR** | **oxidation flow reactor** |
| **VOC** | **volatile organic compound** |
| **SOA** | **secondary organic aerosol** |
| **$H_2O$** | **water vapor mixing ratio** |
| **$OHR_{ext}$** | **external OH reactivity (due to CO, $SO_2$, VOCs etc.)** |
| **PAM** | **Potential Aerosol Mass, a specific type of OFR** |
| **OFR185** | **oxidation flow reactor using both 185 and 254 nm light** |

| | |
|---|---|
| **OFR254** | **oxidation flow reactor using 254 nm light only** |
| **OFR254-X** | **OFR254 with X ppm $O_3$ initially injected** |
| **OFR-i$N_2$O** | **OFR with $N_2$O initially injected** |
| **OFR185-i$N_2$O** | **OFR185 with $N_2$O initially injected** |
| **OFR254-i$N_2$O** | **OFR254 with $N_2$O initially injected** |
| **OFR254-X-i$N_2$O** | **OFR254-X with $N_2$O initially injected** |
| **$OHR_{VOC}$** | **OH reactivity due to VOCs** |
| **F185, F254 etc.** | **UV photon flux at 185 nm, 254 nm etc.** |
| **$N_2$O** | **$N_2$O mixing ratio** |
| **$OH_{exp}$, $F185_{exp}$ etc.** | **exposure (integral over time) to OH, F185 etc.** |

"

R2.4) line 51: remove the extra "(".

We have moved the second "(" in L51 as suggested by the Referee.

R2.5) line 212: I genuinely don't understand this sentence - please clarify (i.e. an accuracy of what?)

We have modified this sentence (in L212) for more clarity. Below is the modified sentence:

"**The outputs of our model (e.g. species concentrations and exposures) were estimated to be accurate to within a factor of 2–3 when compared with field OFR experiments; better agreement can generally be obtained for comparisons with laboratory OFR experiments (Li et al., 2015; Peng et al., 2015).**"

R2.6) line 352: Acyl RO2 +NO2 is typically referred to as an 'equilibrium', not 'quasi-irreversible reaction'. Consider what happens as temperature is increased - in the troposphere in summer, this equilibrium is important for most PAN-type compounds, and cannot be ignored! If this is the case in most OFRs, then there is a more serious problem with the RO2/NO2 and NO/NO2 ratios...

We do not think that a temperature increase of 10–20 K will significantly change the importance of acylperoxy nitrates in OFRs. The equilibrium constant of acyl $RO_2+NO_2\leftrightarrow$acyl $RO_2NO_2$ may change substantially. The O-N bond energy of acylperoxy nitrates is ~28 kcal/mol (Orlando and Tyndall, 2012), which we take as an approximate reaction energy of their decomposition. Then a 20 K temperature increase results in the equilibrium constant shifted toward $RO_2+NO_2$ by x~20. However, this shift is still too small relative to the equilibrium constant itself. For the generic acyl $RO_2$ in this study in an OFR at room temperature (298 K), $RO_2+NO_2\leftrightarrow RO_2NO_2$ has an equilibrium constant $K_1=$~$2x10^{-8}$ $cm^3$ molecule$^{-1}$. In a case with $NO_2$ of $10^{12}$ molecules cm$^{-3}$ (a relatively low level in typical OFR-i$N_2$O experiments; Peng et al., 2018), we set $K_2 = K_1[NO_2]$ = $[RO_2NO_2]/[RO_2]$ = ~$2x10^4$ as the equilibrium constant for $RO_2\leftrightarrow RO_2NO_2$ (only when $[NO_2]$>>$[RO_2]$). Even if reduced by x20 by increasing temperature by 10 K, $K_2$ is still as high as ~1000, which means that only ~1 part per thousand of $RO_2NO_2$ will be present in the reactant form. Even if acylperoxy nitrate decomposition is x20 faster than at room temperature and the formed acyl $RO_2$ can irreversibly react with NO and decrease acylperoxy nitrate concentration, this effect is small: typically up to ~20% decrease in acylperoxy nitrate and usually negligible changes in NO and $NO_2$. The minor effect is due to i) acylperoxy concentration that is still very low, ii) NO concentration that is much lower than $NO_2$ and iii) acylperoxy nitrate decomposition lifetime that is still on the order of minutes.

We believe that it is appropriate to describe acyl $RO_2+NO_2$ in high-$NO_x$ OFRs as a "quasi-irreversible" reaction *at room temperature* and add "**at room temperature**" after "the quasi-irreversible reaction $RO_2+NO_2\rightarrow RO_2NO_2$" in L352 to be more rigorous.

For OFR conditions ~10–20 K higher than room temperature, since they are different than conditions in other reaction systems and that may be unclear to other researchers, we have included a summary of the discussion above in the paper. We have added the new text at the end of the paragraph starting from L583. The added text reads as follows:

"**Besides, reduction of acylperoxy nitrate formation in OFRs, which may be useful to mimic some urban environments where NO plays a larger role in acyl $RO_2$ fate (see Section 3.1.2), is unlikely to be achieved by increasing OFR temperature. The O-N bond energy of acylperoxy nitrates is ~28 kcal/mol (Orlando and Tyndall, 2012), which can be taken as an approximate reaction energy of their decomposition. Then a 20 K temperature increase results in the equilibrium constant of acyl $RO_2+NO_2\leftrightarrow$acyl $RO_2NO_2$ shifted toward $RO_2+NO_2$ by a factor of ~20. However, this shift is still too small relative to the equilibrium constant itself. It can be obtained by a simple calculation that for the generic acyl $RO_2$ in this study in an OFR at 318 K (20 K higher than room temperature) with $NO_2$ of $10^{12}$ molecules cm$^{-3}$ (a relatively low level in typical OFR-i$N_2$O experiments; Peng et al., 2018), ~0.1% of the total amount of acyl $RO_2$ + acyl $RO_2NO_2$ will be present in the form of acyl $RO_2$. Even if acylperoxy nitrate decomposition is x20 faster than at room temperature and the formed acyl $RO_2$ can irreversibly react with NO and decrease acylperoxy nitrate concentration, this effect is small: typically up to ~20% decrease in acylperoxy nitrate and usually negligible changes in NO and $NO_2$. The minor effect is due to i) acylperoxy concentration that is still very low, ii) NO concentration that is much**

**lower than NO₂ and iii) acylperoxy nitrate decomposition lifetime that is still on the order of minutes.**"

R2.7) line 361: what are typical NO/NO2 ratios in the OFR and in the troposphere? It would be helpful to summarize in a sentence.

We have modified the text to L361 to include the information requested by the Referee:

"**RO₂+NO₂ is an inevitable and dominant sink of most acyl RO₂ in high-NOₓ OFRs, though the extent of this dominance differs substantially between the different OFR operation modes. In OFR254-70-iN₂O, RO₂+NO makes a minor or negligible contribution to acyl RO₂ fate because the required high O₃ very rapidly oxidizes NO to NO₂ and leads to very low NO-to-NO₂ ratios (e.g. ~0.003–0.03; see Fig. S7). In OFR185-iN₂O, the contribution of RO₂+NO can be somewhat significant, with typical NO-to-NO₂ of ~0.03–0.4. (Fig S7).Urban NO-to-NO₂ ratios vary widely, for example (roughly, and excluding significant tails in the frequency distributions), 0.02–1 for Barcelona, 0.007–0.7 for Los Angeles and Pittsburgh (see Fig. S7). Given these variations among different urban areas, RO₂+NO and RO₂+NO₂ for acyl RO₂ in OFR185-iN₂O can be regarded as relevant to urban atmospheres. Exceptions to the relevance of OFR185-iN₂O occur during morning rush hours (e.g. see the high NO-to-NO₂ tail for the Pittsburgh case in Fig. S7), near major NO sources, and/or in urban atmospheres with stronger NO emission intensity (e.g. Beijing, especially in winter; Fig. S7). In these cases, NO-to-NO₂ ratios may significantly exceed 1, and RO₂+NO may be the dominant acyl RO₂ loss pathway. Such high-NO conditions appear difficult to simulate in OFRs with the current range of techniques.**

[Figure]

**Figure S7. Frequency occurrence distributions of NO-to-NO$_2$ ratios for OFR185-iN$_2$O and OFR254-70-iN$_2$O model cases and measured at the Los Angeles, Pittsburgh and Barcelona ground sites during the CalNex-LA 2010, PAQS 2002 and DAURE 2009 campaigns, respectively (Zhang et al., 2005; Reche et al., 2011; Ryerson et al., 2013) and at a ground site in Beijing in both summer and winter (Hu et al., 2016). OFR cases under bad conditions are filtered out. The total areas of all distributions are identical.**"

R2.8) line 371: The sentence that states that acyl RO2 dominate aged air plumes requires a reference.

See the response to comment R1.6.

R2.9) line 432: 's' on the end of 'reaction(s)' should be deleted.

We have remove this "s" as suggested by the Referee.

**Additional modification**

We have discovered a bug in the OFR RO$_2$ Fate Estimator that affects the RO$_2$ fate estimation in OFR185 (low-NO mode) and fixed it in the revised Supplement.

**References (for responses to both reviewers)**

Atkinson, R. and Arey, J.: Atmospheric degradation of volatile organic compounds., Chem. Rev., 103(12), 4605–38, doi:10.1021/cr0206420, 2003.

Aumont, B., Szopa, S. and Madronich, S.: Modelling the evolution of organic carbon during its gas-phase tropospheric oxidation: development of an explicit model based on a self generating approach, Atmos. Chem. Phys., 5(9), 2497–2517, doi:10.5194/acp-5-2497-2005, 2005.

Fuchs, H., Novelli, A., Rolletter, M., Hofzumahaus, A., Pfannerstill, E. Y., Kessel, S., Edtbauer, A., Williams, J., Michoud, V., Dusanter, S., Locoge, N., Zannoni, N., Gros, V., Truong, F., Sarda-Esteve, R., Cryer, D. R., Brumby, C. A., Whalley, L. K., Stone, D., Seakins, P. W., Heard, D. E., Schoemaecker, C., Blocquet, M., Coudert, S., Batut, S., Fittschen, C., Thames, A. B., Brune, W. H., Ernest, C., Harder, H., Muller, J. B. A., Elste, T., Kubistin, D., Andres, S., Bohn, B., Hohaus, T., Holland, F., Li, X., Rohrer, F., Kiendler-Scharr, A., Tillmann, R., Wegener, R., Yu, Z., Zou, Q. and Wahner, A.: Comparison of OH reactivity measurements in the atmospheric

simulation chamber SAPHIR, Atmos. Meas. Tech., 10(10), 4023–4053, doi:10.5194/amt-10-4023-2017, 2017.

Hu, W., Hu, M., Hu, W., Jimenez, J. L., Yuan, B., Chen, W., Wang, M., Wu, Y., Chen, C., Wang, Z., Peng, J., Zeng, L. and Shao, M.: Chemical composition, sources, and aging process of submicron aerosols in Beijing: Contrast between summer and winter, J. Geophys. Res. Atmos., 121(4), 1955–1977, doi:10.1002/2015JD024020, 2016.

Huffman, J. A., Docherty, K. S., Aiken, A. C., Cubison, M. J., Ulbrich, I. M., DeCarlo, P. F., Sueper, D., Jayne, J. T., Worsnop, D. R., Ziemann, P. J. and Jimenez, J. L.: Chemically-resolved aerosol volatility measurements from two megacity field studies, Atmos. Chem. Phys., 9(1), 7161–7182, doi:doi:10.5194/acp-9-7161-2009, 2009.

Hunter, J. F., Day, D. A., Palm, B. B., Yatavelli, R. L. N., Chan, A. W. H., Kaser, L., Cappellin, L., Hayes, P. L., Cross, E. S., Carrasquillo, A. J., Campuzano-Jost, P., Stark, H., Zhao, Y., Hohaus, T., Smith, J. N., Hansel, A., Karl, T., Goldstein, A. H., Guenther, A., Worsnop, D. R., Thornton, J. A., Heald, C. L., Jimenez, J. L. and Kroll, J. H.: Comprehensive characterization of atmospheric organic carbon at a forested site, Nat. Geosci., 10(10), 748–753, doi:10.1038/NGEO3018, 2017.

[revised manuscript text omitted]